# Winner Takes It All: Training Performant RL Populations for Combinatorial Optimization

**Nathan Grinsztajn**
InstaDeep
n.grinsztajn@instadeep.com

**Daniel Furelos-Blanco**
Imperial College London*

**Shikha Surana**
InstaDeep

**Clément Bonnet**
InstaDeep

**Thomas D. Barrett**
InstaDeep

## Abstract

Applying reinforcement learning (RL) to combinatorial optimization problems is attractive as it removes the need for expert knowledge or pre-solved instances. However, it is unrealistic to expect an agent to solve these (often NP-)hard problems in a single shot at inference due to their inherent complexity. Thus, leading approaches often implement additional search strategies, from stochastic sampling and beam-search to explicit fine-tuning. In this paper, we argue for the benefits of learning a population of complementary policies, which can be simultaneously rolled out at inference. To this end, we introduce Poppy, a simple training procedure for populations. Instead of relying on a predefined or hand-crafted notion of diversity, Poppy induces an unsupervised specialization targeted solely at maximizing the performance of the population. We show that Poppy produces a set of complementary policies, and obtains state-of-the-art RL results on four popular NP-hard problems: traveling salesman, capacitated vehicle routing, 0-1 knapsack, and job-shop scheduling.

## 1 Introduction

In recent years, machine learning (ML) approaches have overtaken algorithms that use handcrafted features and strategies across a variety of challenging tasks [Mnih et al., 2015, van den Oord et al., 2016, Silver et al., 2018, Brown et al., 2020]. In particular, solving combinatorial optimization (CO) problems – where the maxima or minima of an objective function acting on a finite set of discrete variables is sought – has attracted significant interest [Bengio et al., 2021] due to both their (often NP) hard nature and numerous practical applications across domains varying from logistics [Sbihi and Eglese, 2007] to fundamental science [Wagner, 2020].

As the search space of feasible solutions typically grows exponentially with the problem size, exact solvers can be challenging to scale; hence, CO problems are often also tackled with handcrafted heuristics using expert knowledge. Whilst a diversity of ML-based heuristics have been proposed, reinforcement learning [RL; Sutton and Barto, 2018] is a promising paradigm as it does not require pre-solved examples of these hard problems. Indeed, algorithmic improvements to RL-based CO solvers, coupled with low inference cost, and the fact that they are by design targeted at specific problem distributions, have progressively narrowed the gap with traditional solvers.

RL methods frame CO as sequential decision-making problems, and can be divided into two families [Mazyavkina et al., 2021]. First, *improvement methods* start from a feasible solution and iteratively

---

*Work completed during an internship at InstaDeep.

37th Conference on Neural Information Processing Systems (NeurIPS 2023).

improve it through small modifications (actions). However, such incremental search cannot quickly access very different solutions, and requires handcrafted procedures to define a sensible action space. Second, *construction methods* incrementally build a solution by selecting one element at a time. In practice, it is often unrealistic for a learned heuristic to solve NP-hard problems in a single shot, therefore these methods are typically combined with search strategies, such as stochastic sampling or beam search. However, just as improvement methods are biased by the initial starting solution, construction methods are biased by the single underlying policy. Thus, a balance must be struck between the exploitation of the learned policy (which may be ill-suited for a given problem instance) and the exploration of different solutions (where the extreme case of a purely random policy will likely be highly inefficient).

In this work, we propose Poppy, a construction method that uses a *population* of agents with suitably diverse policies to improve the exploration of the solution space of hard CO problems. Whereas a single agent aims to perform well across the entire problem distribution, and thus has to make compromises, a population can learn a set of heuristics such that only one of these has to be performant on any given problem instance. However, realizing this intuition presents several challenges: (i) naïvely training a population of agents is expensive and challenging to scale, (ii) the trained population should have complementary policies that propose different solutions, and (iii) the training approach should not impose any handcrafted notion of diversity within the set of policies given the absence of clear behavioral markers aligned with performance for typical CO problems.

Challenge (i) can be addressed by sharing a large fraction of the computations across the population, specializing only lightweight policy heads to realize the diversity of agents. Moreover, this can be done on top of a pre-trained model, which we clone to produce the population. Challenges (ii) and (iii) are jointly achieved by introducing an RL objective aimed at specializing agents on distinct subsets of the problem distribution. Concretely, we derive a policy gradient method for the population-level objective, which corresponds to training only the agent which performs best on each problem. This is intuitively justified as the performance of the population on a given problem is not improved by training an agent on an instance where another agent already has better performance. Strikingly, we find that judicious application of this conceptually simple objective gives rise to a population where the diversity of policies is obtained without explicit supervision (and hence is applicable across a range of problems without modification) and essential for strong performance.

Our contributions are summarized as follows:

1. We motivate the use of populations for CO problems as an efficient way to explore environments that are not reliably solved by single-shot inference.

2. We derive a new training objective and present a practical training procedure that encourages performance-driven diversity (i.e. effective diversity without the use of explicit behavioral markers or other external supervision).

3. We evaluate Poppy on four CO problems: traveling salesman (TSP), capacitated vehicle routing (CVRP), 0-1 knapsack (KP), and job-shop scheduling (JSSP). In these four problems, Poppy consistently outperforms all other RL-based approaches.

## 2   Related Work

**ML for Combinatorial Optimization**   The first attempt to solve TSP with neural networks is due to Hopfield and Tank [1985], which only scaled up to 30 cities. Recent developments of bespoke neural architectures [Vinyals et al., 2015, Vaswani et al., 2017] and performant hardware have made ML approaches increasingly efficient. Indeed, several architectures have been used to address CO problems, such as graph neural networks [Dai et al., 2017], recurrent neural networks [Nazari et al., 2018], and attention mechanisms [Deudon et al., 2018]. Kool et al. [2019] proposed an encoder-decoder architecture that we employ for TSP, CVRP and KP. The costly encoder is run once per problem instance, and the resulting embeddings are fed to a small decoder iteratively rolled out to get the whole trajectory, which enables efficient inference. This approach was furthered by Kwon et al. [2020] and Kim et al. [2022], who leveraged the underlying symmetries of typical CO problems (e.g. of starting positions and rotations) to realize improved training and inference performance using instance augmentations. Kim et al. [2021] also draw on Kool et al. and use a hierarchical strategy where a seeder proposes solution candidates, which are refined bit-by-bit by a reviser. Closer to our work, Xin et al. [2021] train multiple policies using a shared encoder and separate decoders.

Whilst this work (MDAM) shares our architecture and goal of training a population, our approach to enforcing diversity differs substantially. MDAM explicitly trades off performance with diversity by jointly optimizing policies and their KL divergence. Moreover, as computing the KL divergence for the whole trajectory is intractable, MDAM is restricted to only using it to drive diversity at the first timestep. In contrast, Poppy drives diversity by maximizing population-level performance (i.e. without any explicit diversity metric), uses the whole trajectory and scales better with the population size (we have used up to 32 agents instead of only 5).

Additionally, ML approaches usually rely on mechanisms to generate multiple candidate solutions [Mazyavkina et al., 2021]. One such mechanism consists in using improvement methods on an initial solution: de O. da Costa et al. [2020] use policy gradients to learn a policy that selects local operators (2-opt) given a current solution in TSP, while Lu et al. [2020] and Wu et al. [2021] extend this method to CVRP. This idea has been extended to enable searching a learned latent space of solutions [Hottung et al., 2021]. However, these approaches have two limitations: they are environment-specific, and the search procedure is inherently biased by the initial solution.

An alternative exploration mechanism is to generate a diverse set of trajectories by stochastically sampling a learned policy, potentially with additional beam search [Joshi et al., 2019], Monte Carlo tree search [Fu et al., 2021], dynamic programming [Kool et al., 2021], active search [Hottung et al., 2022], or simulation-guided search [Choo et al., 2022]. However, intuitively, the generated solutions tend to remain close to the underlying deterministic policy, implying that the benefits of additional sampled candidates diminish quickly.

**Population-Based RL**    Populations have already been used in RL to learn diverse behaviors. In a different context, Gupta et al. [2018], Eysenbach et al. [2019], Hartikainen et al. [2020] and Pong et al. [2020] use a single policy conditioned on a set of goals as an implicit population for unsupervised skill discovery. Closer to our approach, another line of work revolves around explicitly storing a set of distinct policy parameters. Hong et al. [2018], Doan et al. [2020], Jung et al. [2020] and Parker-Holder et al. [2020] use a population to achieve a better coverage of the policy space. However, they enforce explicit attraction-repulsion mechanisms, which is a major difference with respect to our approach where diversity is a pure byproduct of performance optimization.

Our method is also related to approaches combining RL with evolutionary algorithms [EA; Khadka and Tumer, 2018, Khadka et al., 2019, Pourchot and Sigaud, 2019], which benefit from the sample-efficient RL policy updates while enjoying evolutionary population-level exploration. However, the population is a means to learn a unique strong policy, whereas Poppy learns a set of complementary strategies. More closely related, Quality-Diversity [QD; Pugh et al., 2016, Cully and Demiris, 2018] is a popular EA framework that maintains a portfolio of diverse policies. Pierrot et al. [2022] has recently combined RL with a QD algorithm, Map-Elites [Mouret and Clune, 2015]; unlike Poppy, QD methods rely on handcrafted behavioral markers, which is not easily amenable to the CO context.

One of the drawbacks of population-based RL is its expensive cost. However, recent approaches have shown that modern hardware, as well as targeted frameworks, enable efficient vectorized population training [Flajolet et al., 2022], opening the door to a wider range of applications.

## 3    Methods

### 3.1    Background and Motivation

**RL Formulation**    A CO problem instance $\rho$ sampled from some distribution $\mathcal{D}$ consists of a discrete set of $N$ variables (e.g. city locations in TSP). We model a CO problem as a Markov decision process (MDP) defined by a state space $\mathcal{S}$, an action space $\mathcal{A}$, a transition function $T$, and a reward function $R$. A state is a trajectory through the problem instance $\tau_t = (x_1, \ldots, x_t) \in \mathcal{S}$ where $x_i \in \rho$, and thus consists of an ordered list of variables (not necessarily of length $N$). An action, $a \in \mathcal{A} \subseteq \rho$, consists of choosing the next variable to add; thus, given state $\tau_t = (x_1, \ldots, x_t)$ and action $a$, the next state is $\tau_{t+1} = T(\tau_t, a) = (x_1, \ldots, x_t, a)$. Let $\mathcal{S}^* \subseteq \mathcal{S}$ be the set of *solutions*, i.e. states that comply with the problem's constraints (e.g., a sequence of cities such that each city is visited once and ends with the starting city in TSP). The reward function $R : \mathcal{S}^* \to \mathbb{R}$ maps solutions into scalars. We assume the reward is maximized by the optimal solution (e.g. $R$ returns the negative tour length in TSP).

A *policy* $\pi_\theta$ parameterized by $\theta$ can be used to generate solutions for any instance $\rho \sim \mathcal{D}$ by iteratively sampling the next action $a \in \mathcal{A}$ according to the probability distribution $\pi_\theta(\cdot \mid \rho, \tau_t)$. We learn $\pi_\theta$ using REINFORCE [Williams, 1992]. This method aims at maximizing the RL objective $J(\theta) \doteq \mathbb{E}_{\rho \sim \mathcal{D}} \mathbb{E}_{\tau \sim \pi_\theta, \rho} R(\tau)$ by adjusting $\theta$ such that good trajectories are more likely to be sampled in the future. Formally, the policy parameters $\theta$ are updated by gradient ascent using $\nabla_\theta J(\theta) = \mathbb{E}_{\rho \sim \mathcal{D}} \mathbb{E}_{\tau \sim \pi_\theta, \rho}(R(\tau) - b_\rho) \nabla_\theta \log(p_\theta(\tau))$, where $p_\theta(\tau) = \prod_t \pi_\theta(a_{t+1} \mid \rho, \tau_t)$ and $b_\rho$ is a baseline. The gradient of the objective, $\nabla_\theta J$, can be estimated empirically using Monte Carlo simulations.

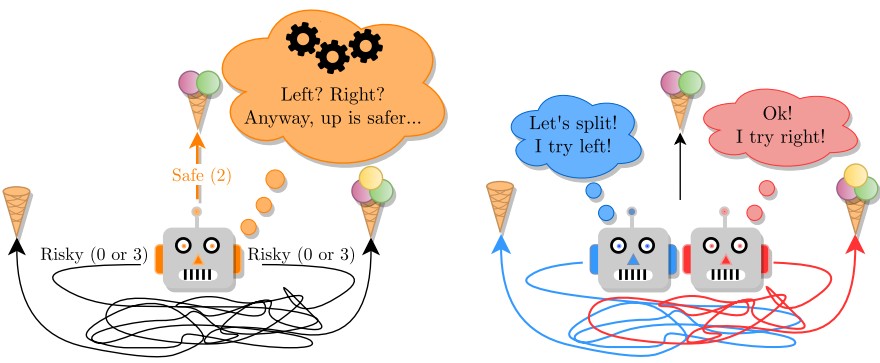

Figure 1: In this environment, the upward path always leads to a medium reward, while the left and right paths are intricate such that either one may lead to a low reward or high reward with equal probability. **Left**: An agent trained to maximize its expected reward converges to taking the safe upward road since acting optimally is too computationally intensive, as it requires solving the maze. **Right**: A 2-agent population can always take the left and right paths and thus get the largest reward.

**Motivating Example**  We argue for the benefits of training a population using the example in Figure 1. In this environment, there are three actions: **Left**, **Right**, and **Up**. **Up** leads to a medium reward, while **Left/Right** lead to low/high or high/low rewards (the configuration is determined with equal probability at the start of each episode). Crucially, the left and right paths are intricate, so the agent cannot easily infer from its observation which one leads to a higher reward. Then, the best strategy for a computationally limited agent is to always go **Up**, as the guaranteed medium reward (2 scoops) is higher than the expected reward of guessing left or right (1.5 scoops). In contrast, two agents in a population can go in opposite directions and always find the maximum reward. There are two striking observations: (i) the agents do not need to perform optimally for the population performance to be optimal (one agent gets the maximum reward), and (ii) the performance of each individual agent is worse than in the single-agent case.

The discussed phenomenon can occur when (i) some optimal actions are too difficult to infer from observations and (ii) choices are irreversible (i.e. it is not possible to recover from a sub-optimal decision). This problem setting can be seen as a toy model for the challenge of solving an NP-hard CO problem in a sequential decision-making setting. In the case of TSP, for example, the number of possible unique tours that could follow from each action is exponentially large and, for any reasonably finite agent capacity, essentially provides the same obfuscation over the final returns. In this situation, as shown above, maximizing the performance of a population will require agents to specialize and likely yield better results than in the single-agent case.

## 3.2 Poppy

We present the components of Poppy: an RL objective encouraging agent specialization, and an efficient training procedure taking advantage of a pre-trained policy.

**Population-Based Objective**  At inference, reinforcement learning methods usually sample several candidates to find better solutions. This process, though, is not anticipated during training, which optimizes the 1-shot performance with the usual RL objective $J(\theta) = \mathbb{E}_{\rho \sim \mathcal{D}} \mathbb{E}_{\tau \sim \pi_\theta, \rho} R(\tau)$ previously presented in Section 3.1. Intuitively, given $K$ trials, we would like to find the best set of policies $\{\pi_1, \dots, \pi_K\}$ to rollout once on a given problem. This gives the following population objective:

$$J_{\text{pop}}(\theta_1, \ldots, \theta_K) \doteq \mathbb{E}_{\rho \sim \mathcal{D}} \mathbb{E}_{\tau_1 \sim \pi_{\theta_1}, \ldots, \tau_K \sim \pi_{\theta_K}} \max \left[ R(\tau_1), \ldots, R(\tau_K) \right],$$

where each trajectory $\tau_i$ is sampled according to the policy $\pi_{\theta_i}$. Maximizing $J_{\text{pop}}$ leads to finding the best set of $K$ agents which can be rolled out in parallel for any problem.

**Theorem 1** (Policy gradient for populations). *The gradient of the population objective is:*

$$\nabla J_{\text{pop}}(\theta_1, \theta_2, \ldots, \theta_K) = \mathbb{E}_{\rho \sim \mathcal{D}} \mathbb{E}_{\tau_1 \sim \pi_{\theta_1}, \ldots, \tau_K \sim \pi_{\theta_K}} \left( R(\tau_{i^*}) - R(\tau_{i^{**}}) \right) \nabla \log p_{\theta_{i^*}}(\tau_{i^*}),$$

*where:* $i^* = \arg\max_{i \in \{1, \ldots, K\}} \left[ R(\tau_i) \right]$ *(index of the agent that got the highest reward) and* $i^{**} = \arg\max_{i \neq i^*} \left[ R(\tau_i) \right]$ *(index of the agent that got the second highest reward).*

The proof is provided in Appendix B.1. Remarkably, it corresponds to rolling out every agent and only training the one that got the highest reward on any problem. This formulation applies across various problems and directly optimizes for population-level performance without explicit supervision or handcrafted behavioral markers.

The theorem trivially holds when instead of a population, one single agent is sampled $K$ times, which falls back to the specific case where $\theta_1 = \theta_2 = \cdots = \theta_K$. In this case, the gradient becomes:

$$\nabla J_{\text{pop}}(\theta) = \mathbb{E}_{\rho \sim \mathcal{D}} \mathbb{E}_{\tau_1 \sim \pi_\theta, \ldots, \tau_K \sim \pi_\theta} \left( R(\tau_{i^*}) - R(\tau_{i^{**}}) \right) \nabla \log p_\theta(\tau_{i^*}),$$

*Remark.* The occurrence of $R(\tau_{i^{**}})$ in the new gradient formulation can be surprising. However, Theorem 1 can be intuitively understood as training each agent on its true contribution to the population's performance: if for any problem instance the best agent $i^*$ was removed, the population performance would fallback to the level of the second-best agent $i^{**}$, hence its contribution is indeed $R(\tau_{i^*}) - R(\tau_{i^{**}})$.

Optimizing the presented objective does not provide any strict diversity guarantee. However, note that diversity maximizes our objective in the highly probable case that, within the bounds of finite capacity and training, a single agent does not perform optimally on all subsets of the training distribution. Therefore, intuitively and later shown empirically, diversity emerges over training in the pursuit of maximizing the objective.

---

**Algorithm 1** Poppy training

---

1: **Input:** problem distribution $\mathcal{D}$, number of agents $K$, batch size $B$, number of training steps $H$, pre-trained parameters $\theta$.
2: $\theta_1, \theta_2, \ldots, \theta_K \leftarrow \texttt{CLONE}(\theta)$ {Clone the pre-trained agent $K$ times.}
3: **for** step 1 to $H$ **do**
4:    $\rho_i \leftarrow \texttt{Sample}(\mathcal{D}) \ \forall i \in 1, \ldots, B$
5:    $\tau_i^k \leftarrow \texttt{Rollout}(\rho_i, \theta_k) \ \forall i \in 1, \ldots, B, \forall k \in 1, \ldots, K$
6:    $k_i^* \leftarrow \arg\max_{k \leq K} R(\tau_i^k) \ \forall i \in 1, \ldots, B$ {Select the best agent for each problem $\rho_i$.}
7:    $\nabla L(\theta_1, \ldots, \theta_K) \leftarrow \frac{1}{B} \sum_{i \leq B} \texttt{REINFORCE}(\tau_i^{k_i^*})$ {Backpropagate through these only.}
8:    $(\theta_1, \ldots, \theta_K) \leftarrow (\theta_1, \ldots, \theta_K) - \alpha \nabla L(\theta_1, \ldots, \theta_K)$

---

**Training Procedure**    The training procedure consists of two phases:

1. We train (or reuse) a single agent using an architecture suitable for solving the CO problem at hand. We later outline the architectures used for the different problems.

2. The agent trained in Phase 1 is cloned $K$ times to form a $K$-agent population. The population is trained as described in Algorithm 1: only the best agent is trained on any problem. Agents implicitly specialize in different types of problem instances during this phase.

Phase 1 enables training the model without the computational overhead of a population. Moreover, we informally note that applying the Poppy objective directly to a population of untrained agents can be unstable. Randomly initialized agents are often ill-distributed, hence a single (or few) agent(s) dominate the performance across all instances. In this case, only the initially dominating agents receive a training signal, further widening the performance gap. Whilst directly training a population

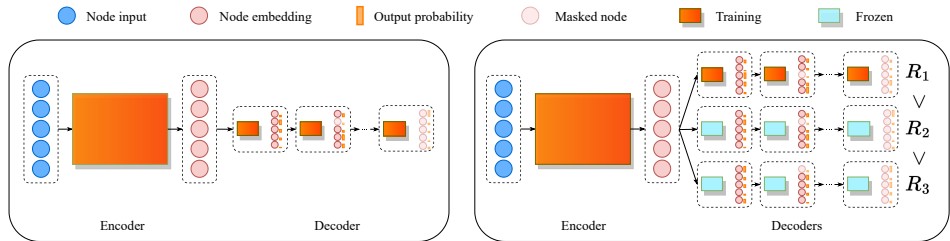

Figure 2: Phases of the training process with a model using static instance embeddings. **Left (Phase 1)**: the encoder and the decoder are trained from scratch. **Right (Phase 2)**: the decoder is cloned $K$ times, and the whole model is trained using the Poppy training objective (i.e. the gradient is only propagated through the decoder that yields the highest reward).

of untrained agents for population-level performance may be achievable with suitable modifications, we instead opt for the described pre-training approach as it is efficient and stable. Throughout this article, we use the reinforcement learning baseline of POMO [Kwon et al., 2020], which was proved to be efficient when training with multiple starting points. We investigate in Appendix C the effect of using the analytical baseline from Theorem 1 instead, which we show leads to improved performance.

**Architecture** To reduce the memory footprint of the population, some of the model parameters can be shared. For example, the architecture for TSP, CVRP, and KP uses the attention model by Kool et al. [2019], which decomposes the policy model into two parts: (i) a large encoder $h_\psi$ that takes an instance $\rho$ as input and outputs embeddings $\omega$ for each of the variables in $\rho$, and (ii) a smaller decoder $q_\phi$ that takes the embeddings $\omega$ and a trajectory $\tau_t$ as input and outputs the probabilities of each possible action. Figure 2 illustrates the training phases of such a model. For JSSP, we implement a similar encoder-decoder architecture taken from Jumanji [Bonnet et al., 2023] with the encoder being shared across the population. For all problems, we build a population of size $K$ by sharing the encoder $h_\psi$ with all agents, and implementing agent-specific decoders $q_{\phi_i}$ for $i \in \{1, \ldots, K\}$. This is motivated by (i) the encoder learning general representations that may be useful to all agents, and (ii) reducing the parameter overhead of training a population by keeping the total number of parameters close to the single-agent case. Please see Appendix A.1 for a discussion on model sizes.

## 4 Experiments

We evaluate Poppy on four CO problems: TSP, CVRP, KP and JSSP. We use the JAX implementations from Jumanji [Bonnet et al., 2023] to leverage hardware accelerators (e.g. TPU). To emphasize the generality of our method, we use the hyperparameters from Kwon et al. [2020] for each problem when possible. We run Poppy for various population sizes to demonstrate its time-performance tradeoffs.

**Training** One training step corresponds to computing policy gradients over a batch of 64 (TSP, CVRP, KP) or 128 (JSSP) randomly generated instances for each agent in the population. Training time varies with problem complexity and training phase. For instance, in TSP with 100 cities, Phase 1 takes 4.5M steps (5 days), whereas Phase 2 takes 400k training steps and lasts 1-4 days depending on the population size. We took advantage of our JAX-based implementation by running all our experiments on a v3-8 TPU.

### 4.1 Routing Problems (TSP, CVRP)

**Tasks** We consider two routing tasks: TSP and CVRP. Given a set of $n$ cities, TSP consists in visiting every city once and coming back to the starting city while minimizing the total traveled distance. CVRP is a variant of TSP where a vehicle with limited capacity departs from a depot node and needs to fulfill the demands of the visited nodes. The vehicle's capacity is restored when the depot is visited.

**Setup** We use an encoder-decoder architecture based on that by Kool et al. [2019] and Kwon et al. [2020] (see Appendix E for details). During Phase 1, an architecture with a single decoder is trained; then, the decoder is cloned for each agent in the population and trained according to Algorithm 1. Following Kwon et al. [2020], we generate multiple solutions for each instance $\rho$ by considering a set of $P \in [1, N]$ *starting points*, where $N$ is the number of instance variables, and use the same reinforce baseline at training. We refer the reader to Appendix A.2 for details.

The training instances are of size $n = 100$ in both tasks. The testing instances for $n = 100$ are due to Kool et al. [2019] and Kwon et al. [2020] for TSP and CVRP, respectively, whereas those for $n \in \{125, 150\}$ are due to Hottung et al. [2022].

**Baselines** We use the exact TSP-solver Concorde Applegate et al. [2006] and the heuristic solver HGS [Vidal et al., 2012, Vidal, 2022] to compute the optimality gaps for TSP and CVRP, respectively. The performance of LKH3 [Helsgaun, 2017] is also reported for TSP. We highlight that the performance of HGS is not necessarily optimal.

In both TSP and CVRP, we evaluate the performance of the RL methods MDAM [Xin et al., 2021] with and without beam search, LIH [Wu et al., 2021], and POMO [Kwon et al., 2020] in three settings: (i) with greedy rollouts, (ii) using $r$ stochastic rollouts, and (iii) using an ensemble of $r$ decoders trained in parallel. The value of $r$ corresponds to the largest tested population (i.e., 16 in TSP and 32 in CVRP). Setting (ii) is run to match Poppy's runtime with its largest population, while Setting (iii) performs Phase 2 without the population objective. We also report the performance of the RL methods Att-GCRN+MCTS [Fu et al., 2021] and 2-Opt-DL [de O. da Costa et al., 2020] in TSP, and NeuRewriter [Chen and Tian, 2019] and NLNS [Hottung and Tierney, 2020] in CVRP.

**Results** Tables 1 and 2 display the results for TSP and CVRP, respectively. The columns show the average tour length, the optimality gap, and the total runtime for each test set. The baseline performances from Fu et al. [2021], Xin et al. [2021], Hottung et al. [2022] and Zhang et al. [2020] were obtained with different hardware (Nvidia GTX 1080 Ti, RTX 2080 Ti, and Tesla V100 GPUs, respectively) and framework (PyTorch); thus, for fairness, we mark these times with ∗. As a comparison guideline, we informally note that these GPU inference times should be approximately divided by 2 to get the converted TPU time.

In both TSP and CVRP, Concorde and HGS remain the best algorithms as they are highly specialized solvers. In relation to RL methods, Poppy reaches the best performance across every performance metric in just a few minutes and, remarkably, performance improves as populations grow.

In TSP, Poppy outperforms Att-GCRN+MCTS despite the latter being known for scaling to larger instances than $n = 100$, hence showing it trades off performance for scale. We also emphasize that specialization is crucial to achieving state-of-the-art performance: Poppy 16 outperforms POMO 16 (ensemble), which also trains 16 agents in parallel but without the population objective (i.e. without specializing to serve as an ablation of the objective).

In CVRP, we observe Poppy 32 has the same runtime as POMO with 32 stochastic rollouts while dividing by 2 the optimal gap for $n = 100$.[2] Interestingly, this ratio increases on the generalization instance sets with $n \in \{125, 150\}$, suggesting that Poppy is more robust to distributional shift.

**Analysis** Figure 3 illustrates the resulting behavior from using Poppy on TSP100. We display on the left the training curves of Poppy for three population sizes: 4, 8, 16. Starting from POMO, the population performances quickly improve. Strikingly, Poppy 4 outperforms POMO with 100 stochastic samples, despite using 25 times fewer rollouts. The rightmost plot shows that whilst the population-level performance improves with population size, the average performance of a random agent from the population on a random instance gets worse. We hypothesize that this is due to agent specialization: when each agent has a narrower target sub-distribution, it learns an even more specialized policy, which is even better (resp. worse) for problem instances in (resp. out) of the target sub-distribution. This is in contrast with a simple policy ensemble, for which the average agent performance would remain the same regardless of the population size. Additional analyses are made in Appendix E.1, where we show that every agent contributes to the population performance. Overall, this illustrates that Poppy agents have learned complementary policies: though they individually

---

[2]A fine-grained comparison between POMO with stochastic sampling and Poppy is in Appendix F.

Table 1: TSP results.

| Method | Inference (10k instances) $n = 100$ | | | 0-shot (1k instances) $n = 125$ | | | $n = 150$ | | |
|---|---|---|---|---|---|---|---|---|---|
| | Obj. | Gap | Time | Obj. | Gap | Time | Obj. | Gap | Time |
| Concorde | 7.765 | 0.000% | 82M | 8.583 | 0.000% | 12M | 9.346 | 0.000% | 17M |
| LKH3 | 7.765 | 0.000% | 8H | 8.583 | 0.000% | 73M | 9.346 | 0.000% | 99M |
| MDAM (greedy) | 7.93 | 2.19% | 36S* | - | - | - | - | - | - |
| MDAM (beam search) | 7.79 | 0.38% | 44M* | - | - | - | - | - | - |
| Att-GCRN+MCTS | - | 0.037% | 15M* | - | - | - | - | - | - |
| 2-Opt-DL | 7.83 | 0.87% | 41M* | - | - | - | - | - | - |
| LIH | 7.87 | 1.42% | 2H* | - | - | - | - | - | - |
| POMO | 7.796 | 0.41% | 5S | 8.635 | 0.61% | 1S | 9.439 | 0.99% | 1S |
| POMO (16 samples) | 7.785 | 0.27% | 1M | 8.619 | 0.42% | 10S | 9.423 | 0.81% | 20S |
| POMO 16 (ensemble) | 7.790 | 0.33% | 1M | 8.629 | 0.53% | 10S | 9.435 | 0.95% | 20S |
| **Poppy 4** | 7.777 | 0.15% | 20S | 8.605 | 0.26% | 3S | 9.389 | 0.46% | 5S |
| **Poppy 8** | 7.772 | 0.09% | 40S | 8.598 | 0.18% | 6S | 9.379 | 0.35% | 10S |
| **Poppy 16** | **7.770** | **0.07%** | 1M | **8.594** | **0.14%** | 10S | **9.372** | **0.27%** | 20S |

Table 2: CVRP results.

| Method | Inference (10k instances) $n = 100$ | | | 0-shot (1k instances) $n = 125$ | | | 0-shot (1k instances) $n = 150$ | | |
|---|---|---|---|---|---|---|---|---|---|
| | Obj. | Gap | Time | Obj. | Gap | Time | Obj. | Gap | Time |
| HGS | 15.56 | 0.000% | 3D | 17.37 | 0.000% | 12H | 19.05 | 0.000% | 16H |
| LKH3 | 15.65 | 0.53% | 6D | 17.50 | 0.75% | 19H | 19.22 | 0.89% | 20H |
| MDAM (greedy) | 16.40 | 5.38% | 45S* | - | - | - | - | - | - |
| MDAM (beam search) | 15.99 | 2.74% | 53M* | - | - | - | - | - | - |
| LIH | 16.03 | 3.00% | 5H* | - | - | - | - | - | - |
| NeuRewriter | 16.10 | 3.45% | 66M* | - | - | - | - | - | - |
| NLNS | 15.99 | 2.74% | 62M* | 18.07 | 4.00% | 9M* | 19.96 | 4.76% | 12M* |
| POMO | 15.87 | 2.00% | 10S | 17.82 | 2.55% | 2S | 19.75 | 3.65% | 3S |
| POMO (32 samples) | 15.77 | 1.31% | 5M | 17.69 | 1.82% | 1M | 19.61 | 2.94% | 1M |
| POMO 32 (ensemble) | 15.78 | 1.36% | 5M | 17.70 | 1.87% | 1M | 19.57 | 2.73% | 1M |
| **Poppy 4** | 15.80 | 1.54% | 1M | 17.72 | 2.01% | 10S | 19.61 | 2.95% | 20S |
| **Poppy 8** | 15.77 | 1.31% | 2M | 17.68 | 1.78% | 20S | 19.58 | 2.75% | 40S |
| **Poppy 32** | **15.73** | **1.06%** | 5M | **17.63** | **1.47%** | 1M | **19.50** | **2.33%** | 1M |

perform worse than the single agent baseline POMO, together they obtain better performance than an ensemble, with every agent contributing to the whole population performance.

## 4.2 Packing Problems (KP, JSSP)

**Tasks** To showcase the versatility of Poppy, we evaluate our method on two packing problems: KP and JSSP. In KP, the goal is to maximize the total value of packed items. JSSP can be seen as a 2D packing problem in which the goal is to minimize the schedule makespan under a set of constraints.

**Setup** For KP, we employ the same architecture and training as for TSP and CVRP. For JSSP, we use an attention-based actor-critic architecture proposed by Bonnet et al. [2023] and use REINFORCE with a critic as a baseline to train our agents. Furthermore, akin to the other problems, we first train the agent until convergence and then duplicate the heads to make our population (see Appendix E for more details). In both problems, the evaluation budget consists of 1,600 samples per instance distributed over the population.

**Baselines** We evaluate our method on KP against several baselines including the optimal solution based on dynamic programming, a greedy heuristic, and POMO [Kwon et al., 2020]. For JSSP, we compare Poppy against the optimal solution obtained by Google OR-Tools [Perron and Furnon, 2019], L2D [Zhang et al., 2020] with greedy rollouts and stochastic sampling (consisting of 8,000 samples

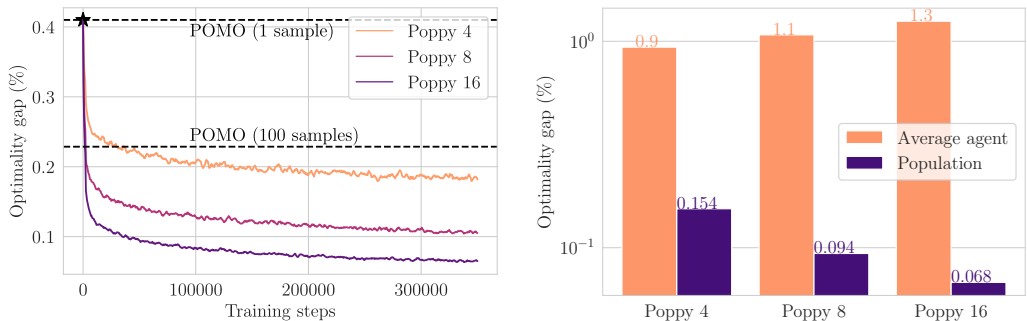

Figure 3: Analysis of Poppy on TSP100. **Left:** Starting from POMO clones, training curves of Poppy for three different population sizes (4, 8, 16). **Right:** With the population objective, the average performance gets worse as the population size increases, but the population-level performance improves.

per problem instance) and our own implementation of MDAM using the same network architecture and budget as Poppy.

**Results**    Table 3 shows the results for KP (3a) and JSSP (3b). The columns indicate the final performance on the test set, which is the average values of items packed for KP and the average schedule duration for JSSP, the optimality gap and the total runtime of these evaluations. The L2D results were obtained on an Intel Core i9-10940X CPU and a single Nvidia GeForce 2080Ti GPU, and thus, their runtimes are marked with ∗.

The results from both packing problems demonstrate that Poppy outperforms other RL baselines. For KP, we observe that Poppy leads to improved performance with a population of 16 agents, dividing the optimality gap with respect to POMO sampling and ensemble by a factor of 12 and 42 for the exact same runtime. We emphasize that although these gaps seem small, these differences are still significant: Poppy 16 is strictly better than POMO in 34.30% of the KP instances, and better in 99.95%. For JSSP, we show that Poppy significantly outperforms L2D in both greedy and sampling settings, and closes the optimality gap by 14.1% and 1.8%, respectively. It should be noted that Poppy outperforms L2D sampling despite requiring 5 times fewer samples. Additionally, to further demonstrate the advantages of training a population of agents, we compare Poppy with a single attention-based model (method named *Single*) evaluated with 16 stochastic samples. The results show that given the same evaluation budget, Poppy outperforms the single agent and closes the optimality gap by a further 1%.

Table 3: Packing problems results.

(a) KP

| Method | Obj. | Gap | Time |
|---|---|---|---|
| | **Testing** (10k instances) $n = 100$ | | |
| Optimal | 40.437 | - | |
| Greedy | 40.387 | 0.1250% | |
| POMO (sampling) | 40.435 | 0.0060% | 2M |
| POMO (ensemble) | 40.429 | 0.021% | 2M |
| **Poppy 16** | **40.437** | **0.0005%** | 2M |

(b) JSSP

| Method | Obj. | Gap | Time |
|---|---|---|---|
| | **Testing** (100 instances) $10 \times 10$ | | |
| OR-Tools (optimal) | 807.6 | 0.0% | 37S |
| L2D (Greedy) | 988.6 | 22.3% | 20S* |
| L2D (Sampling) | 871.7 | 8.0% | 8H* |
| MDAM | 875.0 | 8.3% | 40M |
| Single (16 samples) | 866.0 | 7.2% | 30M |
| **Poppy 16** | **857.7** | **6.2%** | 30M |

## 5    Conclusions

Poppy is a population-based RL method for CO problems. It uses an RL objective that incurs agent specialization with the purpose of maximizing population-level performance. Crucially, Poppy does

not rely on handcrafted notions of diversity to enforce specialization. We show that Poppy achieves state-of-the-art performance on four popular NP-hard problems: TSP, CVRP, KP and JSSP.

This work opens the door to several directions for further investigation. Firstly, we have experimented on populations of at most 32 agents; therefore, it is unclear what the consequences of training larger populations are. Whilst even larger populations could reasonably be expected to provide stronger performance, achieving this may not be straightforward. Aside from the increased computational burden, we also hypothesize that the population performance could eventually collapse once no additional specialization niches can be found, leaving agents with null contributions behind. Exploring strategies to scale and prevent such collapses is an interesting direction for future work.

Secondly, our work has built on the current state-of-the-art RL for CO approaches in a single- or few-shot inference setting to demonstrate the remarkable efficacy of a population-based approach. However, there are other paradigms that we could consider. For example, active-search methods allow an increased number of solving attempts per problem and, in principle, such methods for inference-time adaption of the policy could be combined with an initially diverse population to further boost performance. Indeed, we investigate the performance of Poppy with a larger time budget in Appendix D and find that Poppy combined with a simple sampling procedure and no fine-tuning matches, or even surpasses, the state-of-the-art active search approach of Hottung et al. [2022].

Finally, we recall that the motivation behind Poppy was dealing with problems where predicting optimal actions from observations is too difficult to be solved reliably by a single agent. We believe that such settings are not strictly limited to canonical CO problems, and that population-based approaches offer a promising direction for many challenging RL applications. Specifically, Poppy is amenable in settings where (i) there is a distribution of problem instances and (ii) these instances can be attempted multiple times at inference. These features encompass various fields such as code and image generation, theorem proving, and protein design. In this direction, we hope that approaches that alleviate the need for handcrafted behavioral markers whilst still realizing performant diversity akin to Poppy, could broaden the range of applications of population-based RL.

## Acknowledgements

Research supported with Cloud TPUs from Google's TPU Research Cloud (TRC).

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

# Appendix

## Table of Contents

---
**Algorithm 2** Poppy training with starting points
---
1: **Input:** problem distribution $\mathcal{D}$, number of starting points per instance $P$, number of agents $K$, batch size $B$, number of training steps $H$, a pretrained encoder $h_\psi$ and decoder $q_\phi$
2: $\phi_1, \phi_2, \ldots, \phi_K \leftarrow \texttt{CLONE}(\phi)$ {Clone the pre-trained decoder parameters $K$ times.}
3: **for** step 1 to $H$ **do**
4: $\quad \rho_i \leftarrow \texttt{Sample}(\mathcal{D}) \; \forall i \in 1, \ldots, B$
5: $\quad \alpha_{i,1}, \ldots, \alpha_{i,P} \leftarrow \texttt{SelectStartPoints}(\rho_i, P) \; \forall i \in 1, \ldots, B$
6: $\quad \tau_{i,p}^k \leftarrow \texttt{Rollout}(\rho_i, \alpha_{i,p}, h_\psi, q_{\phi_k}) \; \forall i \in 1, \ldots, B, \forall p \in 1, \ldots, P, \forall k \in 1, \ldots, K$
7: $\quad b_i^k \leftarrow \frac{1}{P} \sum_p R(\tau_{i,p}^k)$
8: $\quad k_{i,p}^* \leftarrow \arg\max_{k \le K} R(\tau_{i,p}^k) \; \forall i \in 1, \ldots, B, \forall p \in 1, \ldots, P$ {Select the best agent per (instance, starting point).}
9: $\quad \nabla L(h_\psi, q_{\phi_1}, q_{\phi_2}, \ldots, q_{\phi_K}) \leftarrow -\frac{1}{BP} \sum_{i,p} (R(\tau_{i,p}^{k_{i,p}^*}) - b_i^{k_{i,p}^*}) \nabla \log p_{\psi, \phi_{k_{i,p}^*}}(\tau_{i,p}^{k_{i,p}^*})$ {Propagate gradients through these only.}
10: $\quad (h_\psi, q_{\phi_1}, q_{\phi_2}, \ldots, q_{\phi_K}) \leftarrow (h_\psi, q_{\phi_1}, q_{\phi_2}, \ldots, q_{\phi_K}) - \alpha \nabla L(h_\psi, q_{\phi_1}, q_{\phi_2}, \ldots, q_{\phi_K})$
---

# A   Additional Details on Poppy

## A.1   Number of Parameters (TSP)

Table 4 shows the total number of parameters of our models as a function of the population size when experimenting on TSP. Since the decoder represents less than $10\%$ of the parameters, scaling the population size can be done efficiently. For instance, a population of 16 agents roughly doubles the model size. This observation transfers to CVRP and KP whose encoder-decoder architectures are similar to TSP. The architecture for JSSP is slightly different but the observation remains that the decoder represents a smaller part of the model ( 18%) and thus the population can be efficiently scaled.

Table 4: Number of parameters for different population sizes.

|  | Encoder | Decoder | Population size | | | | |
|---|---|---|---|---|---|---|---|
|  |  |  | 1 | 4 | 8 | 16 | 32 |
| Parameters | 1,190,016 | 98,816 | 1,288,832 | 1,585,280 | 1,980,544 | 2,771,072 | 4,352,128 |
| Extra parameters | - | - | 0% | 23% | 54% | 115% | 238% |

## A.2   Training Details

In Section 3.2 (see "Training Procedure"), we described that Poppy consists of two phases. In a nutshell, the first phase consists of training our model in a single-agent setting (i.e., an encoder-decoder model with a single decoder head), whereas the second phase consists of keeping the encoder and cloning the previously trained decoder $K$ times (where $K$ is the number of agents) and specialize them using the population objective. Algorithm 2 shows the low-level implementation details of the training of the population (i.e., Phase 2) for environments where POMO [Kwon et al., 2020] uses several starting points; namely, given $K$ agents and $P$ starting points, $P \times K$ trajectories are rolled out for each instance, among which only $P$ are effectively used for training.

Every approach was trained until convergence. Given TSP, CVRP, KP and JSSP, the single-agent baselines were trained for 5 days, 7 days, 1 day, and 1 day respectively, while the largest populations we provide were trained for 4 days, 5 days, 3 days, and 3 days respectively. We emphasize that Poppy is already performant with less training budget; indeed, Figure 3 (left) shows that the optimality gap of Poppy decreases very quickly at the beginning, although it takes time to reach convergence. Concretely, only a few hours of training is sufficient for Poppy 4 to reach the performance of POMO with 100 samples.

# B    Mathematical Elements

## B.1    Gradient derivation

We recall that the population objective for $K$ is defined as:

$$J_{\text{pop}}(\theta_1, \theta_2, \ldots, \theta_n) \doteq \mathbb{E}_{\rho \sim \mathcal{D}} \mathbb{E}_{\tau_1 \sim \pi_{\theta_1}, \ldots, \tau_K \sim \pi_{\theta_K}} \max\left[R(\tau_1), \ldots, R(\tau_K)\right].$$

**Theorem** (Policy gradient for the population objective). *The gradient of the population objective is given by:*

$$\nabla J_{\text{pop}}(\theta_1, \theta_2, \ldots, \theta_n) = \mathbb{E}_{\rho \sim \mathcal{D}} \mathbb{E}_{\tau_1 \sim \pi_{\theta_1}, \ldots, \tau_K \sim \pi_{\theta_K}} \left(R(\tau_{i^*}) - R(\tau_{i^{**}})\right) \nabla \log p_{\theta_{i^*}}(\tau_{i^*}),$$

*where:*

$$i^* = \arg\max\left[R(\tau_1), \ldots, R(\tau_K)\right], \qquad \textit{(index of the best trajectory)}$$

$$i^{**} = \arg\text{second}\max\left[R(\tau_1), \ldots, R(\tau_K)\right], \qquad \textit{(index of the second best trajectory)}$$

*Proof.* We first derive the gradient with respect to $\theta_1$ for convenience. As all the agents play a symmetrical role in the objective, the same procedure can be applied to any index.

$$\nabla_{\theta_1} J_{\text{pop}}(\theta_1, \theta_2, \ldots, \theta_K) = \nabla_{\theta_1} \mathbb{E}_{\rho \sim \mathcal{D}} \mathbb{E}_{\tau_1 \sim \pi_{\theta_1}, \ldots, \tau_K \sim \pi_{\theta_K}} \max_{i \in \{1,2,\ldots,K\}} \left[R(\tau_i)\right]$$

$$= \mathbb{E}_{\rho \sim \mathcal{D}} \mathbb{E}_{\tau_1, \ldots, \tau_K} \max_{i \in \{1,2,\ldots,K\}} \left[R(\tau_i)\right] \nabla_{\theta_1} \log p(\tau_1, \ldots, \tau_K)$$

$$= \mathbb{E}_{\rho \sim \mathcal{D}} \mathbb{E}_{\tau_1, \ldots, \tau_K} \max_{i \in \{1,2,\ldots,K\}} \left[R(\tau_i)\right] \nabla_{\theta_1} \log(\pi_{\theta_1}(\tau_1) \ldots \pi_{\theta_K}(\tau_K))$$

$$= \mathbb{E}_{\rho \sim \mathcal{D}} \mathbb{E}_{\tau_1, \ldots, \tau_K} \max_{i \in \{1,2,\ldots,K\}} \left[R(\tau_i)\right] \nabla_{\theta_1} (\log \pi_{\theta_1}(\tau_1) + \cdots + \log \pi_{\theta_K}(\tau_K))$$

$$= \mathbb{E}_{\rho \sim \mathcal{D}} \mathbb{E}_{\tau_1, \ldots, \tau_K} \max_{i \in \{1,2,\ldots,K\}} \left[R(\tau_i)\right] \nabla_{\theta_1} \log \pi_{\theta_1}(\tau_1),$$

We also have for any problem instance $\rho$ and any trajectories $\tau_2, \ldots, \tau_K$:

$$\mathbb{E}_{\tau_1 \sim \pi_{\theta_1}} \max_{i \in \{2,\ldots,K\}} \left[R(\tau_i)\right] \nabla_{\theta_1} \log \pi_{\theta_1}(\tau_1) = \max_{i \in \{2,\ldots,K\}} \left[R(\tau_i)\right] \mathbb{E}_{\tau_1 \sim \pi_{\theta_1}} \nabla_{\theta_1} \log \pi_{\theta_1}(\tau_1)$$

$$= \max_{i \in \{2,\ldots,K\}} \left[R(\tau_i)\right] \mathbb{E}_{\tau_1 \sim \pi_{\theta_1}} \frac{\nabla_{\theta_1} \pi_{\theta_1}(\tau_1)}{\pi_{\theta_1}(\tau_1)}$$

$$= \max_{i \in \{2,\ldots,K\}} \left[R(\tau_i)\right] \sum_{\tau_1} \nabla_{\theta_1} \pi_{\theta_1}(\tau_1)$$

$$= \max_{i \in \{2,\ldots,K\}} \left[R(\tau_i)\right] \nabla_{\theta_1} \sum_{\tau_1} \pi_{\theta_1}(\tau_1)$$

$$= \max_{i \in \{2,\ldots,K\}} \left[R(\tau_i)\right] \nabla_{\theta_1} 1 = 0$$

Intuitively, $\max_{i \in \{2,\ldots,K\}} \left[R(\tau_i)\right]$ does not depend on the first agent, so this derivation simply shows that $\max_{i \in \{2,\ldots,K\}} \left[R(\tau_i)\right]$ can be used as a baseline for training $\theta_1$.

Subtracting this to the quantity obtained in Equation B.1, we have:

$$\nabla_{\theta_1} J_{\text{pop}}(\theta_1, \theta_2, \ldots, \theta_K) = \mathbb{E}_{\rho \sim \mathcal{D}} \mathbb{E}_{\tau_1, \ldots, \tau_K} \max_{i \in \{1,2,\ldots,K\}} \big[ R(\tau_i) \big] \nabla_{\theta_1} \log \pi_{\theta_1}(\tau_1),$$

$$= \mathbb{E}_{\rho \sim \mathcal{D}} \mathbb{E}_{\tau_2, \ldots, \tau_K} \mathbb{E}_{\tau_1} \max_{i \in \{1,2,\ldots,K\}} \big[ R(\tau_i) \big] \nabla_{\theta_1} \log \pi_{\theta_1}(\tau_1),$$

$$= \mathbb{E}_{\rho \sim \mathcal{D}} \mathbb{E}_{\tau_2, \ldots, \tau_K} \mathbb{E}_{\tau_1} \left( \max_{i \in \{1,2,\ldots,K\}} \big[ R(\tau_i) \big] - \max_{i \in \{2,\ldots,K\}} \big[ R(\tau_i) \big] \right) \nabla_{\theta_1} \log \pi_{\theta_1}(\tau_1),$$

$$= \mathbb{E}_{\rho \sim \mathcal{D}} \mathbb{E}_{\tau_2, \ldots, \tau_K} \mathbb{E}_{\tau_1} \mathbb{1}_{i^*=1} \big( R(\tau_1) - R(\tau_{i^{**}}) \big) \nabla_{\theta_1} \log \pi_{\theta_1}(\tau_1),$$

$$= \mathbb{E}_{\rho \sim \mathcal{D}} \mathbb{E}_{\tau_1, \ldots, \tau_K} \mathbb{1}_{i^*=1} \big( R(\tau_{i^*}) - R(\tau_{i^{**}}) \big) \nabla_{\theta_1} \log \pi_{\theta_1}(\tau_1).$$

Equation (B.1) comes from the fact that $\big( \max_{i \in \{1,2,\ldots,K\}} \big[ R(\tau_i) \big] - \max_{i \in \{2,\ldots,K\}} \big[ R(\tau_i) \big] \big)$ is 0 if the best trajectory is not $\tau_1$, and $R(\tau_1) - \max_{i \in \{2,\ldots,K\}} \big[ R(\tau_i) \big] = R(\tau_1) - R(\tau_{i^{**}})$ otherwise.

Finally, for any $j \in \{1, \ldots, K\}$, the same derivation gives:

$$\nabla_{\theta_j} J_{\text{pop}}(\theta_1, \theta_2, \ldots, \theta_K) = \mathbb{E}_{\rho \sim \mathcal{D}} \mathbb{E}_{\tau_1, \ldots, \tau_K} \mathbb{1}_{i^*=j} \big( R(\tau_{i^*}) - R(\tau_{i^{**}}) \big) \nabla_{\theta_j} \log \pi_{\theta_j}(\tau_j).$$

Therefore, we have:

$$\nabla_\theta = \sum_{j=1}^{n} \mathbb{E}_{\rho \sim \mathcal{D}} \mathbb{E}_{\tau_1, \ldots, \tau_K} \mathbb{1}_{i^*=j} \big( R(\tau_{i^*}) - R(\tau_{i^{**}}) \big) \nabla_{\theta_j} \log \pi_{\theta_j}(\tau_j),$$

$$\nabla_\theta = \mathbb{E}_{\rho \sim \mathcal{D}} \mathbb{E}_{\tau_1, \ldots, \tau_K} \big( R(\tau_{i^*}) - R(\tau_{i^{**}}) \big) \nabla_{\theta_{i^*}} \log \pi_{\theta_{i^*}}(\tau_{i^*}),$$

which concludes the proof. □

## C  Baseline Impact Analysis

The results presented in the paper use the same reinforcement learning baseline as POMO [Kwon et al., 2020], the average reward over starting points, as presented in Alg. 2. However, Theorem 1 directly provides an alternative baseline: the reward of the second-best agent. Figures 5–7 compare the performance of Poppy using the baseline from POMO against Poppy using the baseline analytically computed in Theorem 1.

In almost all cases, we observe that the analytical baseline provides better or equal performance than the POMO baseline. We believe that this highlights both that our intuitive "winner takes it all" approach works well even with slightly different choices of baseline, and that indeed our theoretical analysis correctly predicts a modified baseline that can be used in practice with strong performance.

Table 5: TSP results.

| Method | **Inference** (10k instances) | | | **0-shot** (1k instances) | | | | | |
| | $n = 100$ | | | $n = 125$ | | | $n = 150$ | | |
| | Obj. | Gap | Time | Obj. | Gap | Time | Obj. | Gap | Time |
| Concorde | 7.765 | 0.000% | 82M | 8.583 | 0.000% | 12M | 9.346 | 0.000% | 17M |
| LKH3 | 7.765 | 0.000% | 8H | 8.583 | 0.000% | 73M | 9.346 | 0.000% | 99M |
| POMO 16 (ensemble) | 7.790 | 0.33% | 1M | 8.629 | 0.53% | 10S | 9.435 | 0.95% | 20S |
| Poppy 16 (POMO baseline) | 7.770 | 0.07% | 1M | 8.594 | 0.14% | 10S | **9.372** | **0.27%** | 20S |
| Poppy 16 (analytical baseline) | **7.769** | **0.05%** | 1M | **8.594** | **0.13%** | 10S | 9.372 | 0.28% | 20S |

Table 6: CVRP results.

| Method | Inference (10k instances) $n = 100$ | | | 0-shot (1k instances) $n = 125$ | | | 0-shot (1k instances) $n = 150$ | | |
|---|---|---|---|---|---|---|---|---|---|
| | Obj. | Gap | Time | Obj. | Gap | Time | Obj. | Gap | Time |
| LKH3 | 15.65 | 0.000% | 6D | 17.50 | 0.000% | 19H | 19.22 | 0.000% | 20H |
| POMO 32 (ensemble) | 15.78 | 0.83% | 5M | 17.70 | 1.11% | 1M | 19.57 | 1.83% | 1M |
| Poppy 32 (POMO baseline) | 15.73 | 0.52% | 5M | **17.63** | **0.71%** | 1M | **19.50** | **1.43%** | 1M |
| Poppy 32 (analytical baseline) | **15.72** | **0.48%** | 5M | **17.63** | **0.71%** | 1M | **19.50** | **1.43%** | 1M |

Table 7: Packing problems results.

(a) KP

| Method | Testing (10k instances) $n = 100$ | | |
|---|---|---|---|
| | Obj. | Gap | Time |
| Optimal | 40.437 | - | |
| Greedy | 40.387 | 0.1250% | |
| POMO 16 (ensemble) | 40.429 | 0.021% | 2M |
| Poppy 16 (POMO baseline) | 40.437 | 0.0005% | 2M |
| Poppy 16 (analytical baseline) | **40.437** | **0.0001%** | 2M |

(b) JSSP

| Method | Testing (100 instances) $10 \times 10$ | | |
|---|---|---|---|
| | Obj. | Gap | Time |
| OR-Tools (optimal) | 807.6 | 0.0% | 37S |
| Single (16 samples) | 866.0 | 7.2% | 30M |
| Poppy 16 (POMO baseline) | 857.7 | 6.2% | 30M |
| Poppy 16 (analytical baseline) | **852.2** | **5.5%** | 30M |

# D    Comparison to Active Search

We implement a simple sampling strategy to give a sense of the performance of Poppy with a larger time budget. Given a population of $K$ agents, we first greedily rollout each of them on every starting point, and evenly distribute any remaining sampling budget across the most promising $K$ (agent, starting point) pairs for each instance with stochastic rollouts. This two-step process is motivated by the idea that is not useful to sample several times an agent on an instance where it is outperformed by another one. For environments without starting points like JSSP, we stick to the simplest approach of evenly distributing the rollouts across the population, although better performance could likely be obtained by selectively assigning more budget to the best agents.

**Setup**    For TSP, CVRP and JSSP, we use the same test instances as in Tables 1, 2 and 3b. For TSP and CVRP, we generate a total of $200 \times 8 \times N$ candidate solutions per instance (where 8 corresponds to the augmentation strategy by Kwon et al. [2020] and $N$ is the number of starting points), accounting for both the first and second phases. We evaluate our approach against POMO [Kwon et al., 2020], EAS [Hottung et al., 2022] with the same budget, and SGBS [Choo et al., 2022], as well as against the supervised methods GCN-BS [Joshi et al., 2019], CVAE-Opt [Hottung et al., 2021], and DPDP [Kool et al., 2021]. As EASand SGBS have different variants, we compare against those that do not require backpropagating gradients through the network, similarly to our approach; thus, it should match Poppy's compute time on the same hardware. We additionally compare against the evolutionary approach HGS [Vidal et al., 2012, Vidal, 2022]. For JSSP, we use the same setting as EAS [Hottung et al., 2022], and sample a total of 8,000 solutions per problem instance for each approach. For a proper comparison, we reimplemented EAS with the same model architecture as Poppy.

**Results**    Tables 8, 9 and 10 show the results for TSP, CVRP and JSSP respectively. With extra sampling, Poppy reaches a performance gap of $0.002\%$ on TSP100, and establishes a state-of-the-art for general ML-based approaches, even when compared to supervised methods. For CVRP, adding sampling to Poppy makes it on par with DPDP and EAS, depending on the problem size, and it is only outperformed by the active search approach EAS, which gives large improvements on CVRP. As the two-step sampling process used for Poppy is very rudimentary compared to the active search method described in Hottung et al. [2022], it is likely that combining the two approaches could further boost performance, which we leave for future work.

Table 8: TSP results (active search)

| | Method | Inference (10k instances) n = 100 | | | 0-shot (1k instances) n = 125 | | | n = 150 | | |
|---|---|---|---|---|---|---|---|---|---|---|
| | | Obj. | Gap | Time | Obj. | Gap | Time | Obj. | Gap | Time |
| | Concorde | 7.765 | 0.000% | 82M | 8.583 | 0.000% | 12M | 9.346 | 0.000% | 17M |
| | LKH3 | 7.765 | 0.000% | 8H | 8.583 | 0.000% | 73M | 9.346 | 0.000% | 99M |
| SL | GCN-BS | 7.87 | 1.39% | 40M* | - | - | - | - | - | - |
| | CVAE-Opt | - | 0.343% | 6D* | 8.646 | 0.736% | 21H* | 9.482 | 1.45% | 30H* |
| | DPDP | 7.765 | 0.004% | 2H* | 8.589 | 0.070% | 31M* | 9.434 | 0.94% | 44M* |
| RL | POMO (200 samples) | 7.769 | 0.056% | 2H | 8.594 | 0.13% | 20M | 9.376 | 0.31% | 32M |
| | SGBS | 7.769 | 0.058% | 9M* | - | - | - | 9.367 | 0.22% | 8M |
| | EAS | 7.768 | 0.048% | 5H* | 8.591 | 0.091% | 49M* | 9.365 | 0.20% | 1H* |
| | **Poppy 16 (200 samples)** | **7.765** | **0.002%** | 2H | **8.584** | **0.009%** | 20M | **9.351** | **0.05%** | 32M |

Table 9: CVRP results (active search)

| | Method | Inference (10k instances) n = 100 | | | 0-shot (1k instances) n = 125 | | | 0-shot (1k instances) n = 150 | | |
|---|---|---|---|---|---|---|---|---|---|---|
| | | Obj. | Gap | Time | Obj. | Gap | Time | Obj. | Gap | Time |
| | HGS | 15.56 | 0.00% | 3D | 17.37 | 0.00% | 12H | 19.05 | 0.00% | 16H |
| | LKH3 | 15.65 | 0.53% | 6D | 17.50 | 0.75% | 19H | 19.22 | 0.89% | 20H |
| SL | CVAE-Opt | - | 1.90% | 11D* | 17.87 | 2.85% | 36H* | 19.84 | 4.16% | 46H* |
| | DPDP | 15.63 | 0.40% | 23H* | 17.51 | 0.82% | 3H* | 19.31 | 1.38% | 5H* |
| RL | POMO (200 samples) | 15.67 | 0.71% | 4H | 17.56 | 1.08% | 43M | 19.43 | 1.98% | 1H |
| | SGBS | 15.66 | 0.61% | 10M* | - | - | - | 19.43 | 1.96% | 4M* |
| | EAS | **15.62** | **0.39%** | 8H* | 17.49 | 0.75% | 80M* | 19.36 | 1.62% | 2H* |
| | **Poppy 32 (200 samples)** | **15.62** | **0.39%** | 4H | **17.49** | **0.65%** | 42M | **19.32** | **1.40%** | 1H |

# E  Problems

We here describe the details of the four CO problems we have used to evaluate Poppy, namely TSP, CVRP, KP and JSSP. We use the corresponding implementations from Jumanji [Bonnet et al., 2023]: TSP, CVRP, Knapsack and JobShop. For each problem, we describe below the training (e.g. architecture, hyperparameters) and the process of instance generation. In addition, we show some example solutions obtained by a population of agents on TSP and CVRP. Finally, we thoroughly analyze the performance of the populations in TSP.

## E.1  Traveling Salesman Problem (TSP)

**Instance Generation**  The $n$ cities that constitute each problem instance have their coordinates uniformly sampled from $[0, 1]^2$.

**Architecture**  We use the same model as Kool et al. [2019] and Kwon et al. [2020] except for the batch-normalization layers, which are replaced by layer-normalization to ease parallel batch processing. We invert the mask used in the decoder computations (i.e., masking the available cities instead of the unavailable ones) after experimentally observing faster convergence rates. The results

Table 10: JSSP

| Method | Inference (100 instances) 10 × 10 | | |
|---|---|---|---|
| | Obj. | Gap | Time |
| OR-Tools (optimal) | 807.6 | 0.0% | 37S |
| L2D (Greedy) | 988.6 | 22.3% | 20S* |
| L2D (Sampling) | 871.7 | 8.0% | 8H* |
| EAS-L2D | 860.2 | 6.5% | 8H* |
| Sampling | 862.1 | 6.7% | 3H |
| EAS | 858.4 | 6.3% | 3H |
| **Poppy 16** | **849.7** | **5.2%** | 3H |

reported for POMO were obtained with the same implementation changes to keep the comparison fair. These results are on par with those reported in POMO [Kwon et al., 2020].

**Hyperparameters** To match the setting used by Kwon et al. [2020], we use the Adam optimizer [Kingma and Ba, 2015] with a learning rate $\mu = 10^{-4}$, and an $L_2$ penalization coefficient of $10^{-6}$. The encoder is composed of 6 multi-head self-attention layers with 8 heads each. The dimension of the keys, queries and values is 16. Each attention layer is composed of a feed-forward layer of size 512, and the final node embeddings have a dimension of 128. The decoders are composed of 1 multi-head attention layer with 8 heads and 16-dimensional key, query and value.

The number of starting points $P$ is 50 for each instance. We determined this value after performing a grid-search based on the first training steps with $P \in \{20, 50, 100\}$.

**Example Solutions** Figure 5 shows some trajectories obtained from a 16-agent population on TSP100. Even though they look similar, small decisions differ between agents, thus frequently leading to different solutions. Interestingly, some agents (especially 6 and 11) give very poor trajectories. We hypothesize that it is a consequence of specializing since agents have no incentive to provide a good solution if another agent is already better on this instance.

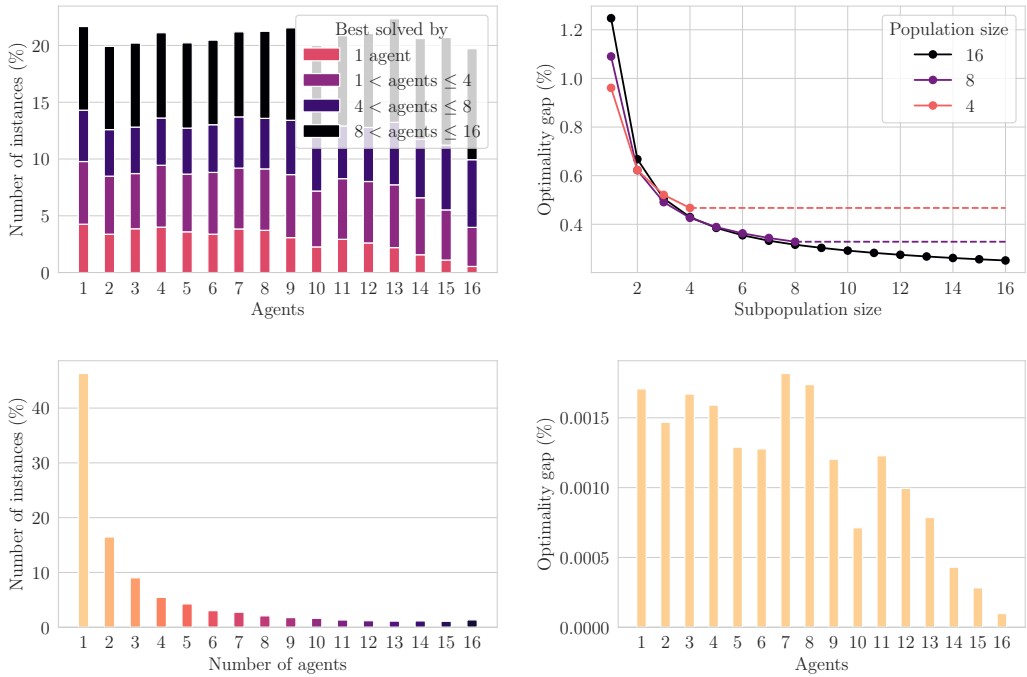

Figure 4: **Upper left**: Proportion of instances that each agent solves best among the population for Poppy 16 on TSP100. Colors indicate the number of agents in the population giving the same solution for these sets of instances. **Upper right**: The mean performance of 1,000 randomly drawn sub-populations for Poppy 1, 4, 8 and 16. **Bottom left**: Proportion of test instances where any number of Poppy 16 agents reaches the exact same best solution. The best performance is reached by only a single agent in 47% of the cases. **Bottom Right**: Optimality gap loss suffered when removing any agent from the population using Poppy 16. Although some agents contribute more (e.g. 7, 8) and some less (e.g. 15, 16), the distribution is relatively even, even though no explicit mechanism enforces this behavior

**Population Analysis** Figure 4 shows some additional information about individual agent performances. In the left figure, we observe that each agent gives on average the best solution for 20% of the instances, and that for around 4% it gives the unique best solution across the population. These numbers are generally evenly distributed, which shows that every agent contributes to the whole population performance. Furthermore, we observe the performance is quite evenly distributed

across the population of Poppy 16; hence, showing that the population has not collapsed to a few high-performing agents, and that Poppy benefits from the population size, as shown in the bottom figure. On the right is displayed the performance of several sub-populations of agents for Poppy 4, 8 and 16. Unsurprisingly, fixed size sub-populations are generally better when sampled from smaller populations: one or two agents randomly sampled from Poppy 4 perform better than when sampled from Poppy 8 or Poppy 16. This highlights the fact that agents have learned complementary behaviors which might be sub-optimal if part of the total population is missing, and this effect is stronger for large populations.

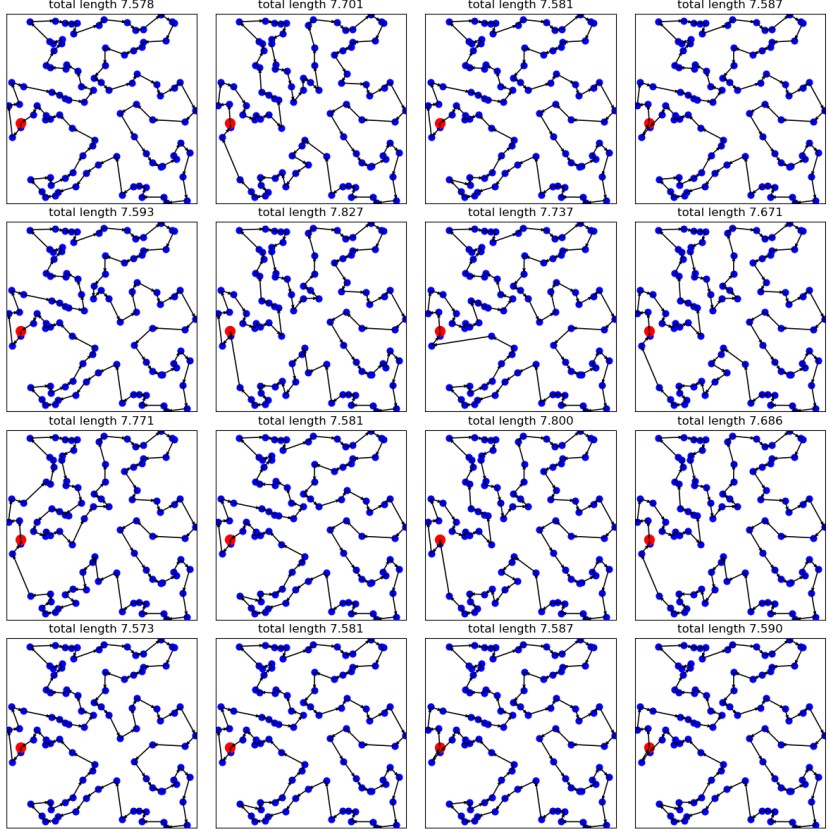

Figure 5: Example TSP trajectories given by Poppy for a 16-agent population from one starting point (red).

## E.2    Capacitated Vehicle Routing Problem (CVRP)

**Instance Generation**    The coordinates of the $n$ customer nodes and the depot are uniformly sampled in $[0, 1]^2$. The demands are uniformly sampled from the discrete set $\{\frac{1}{D}, \frac{2}{D}, \ldots, \frac{9}{D}\}$ where $D = 50$ for CVRP100, $D = 55$ for CVRP125, and $D = 60$ for CVRP150. The maximum vehicle capacity is 1. The deliveries cannot be split: each customer node is visited once, and its whole demand is taken off the vehicle's remaining capacity.

**Architecture**    We use the same model as in TSP. However, unlike TSP, the mask is not inverted; besides, it does not only prevent the agent from revisiting previous customer nodes, but also from

visiting the depot if it is the current location, and any customer node whose demand is higher than the current capacity.

**Hyperparameters**   We use the same hyperparameters as in TSP except for the number of starting points $P$ per instance used during training, which we set to 100 after performing a grid-search with $P \in \{20, 50, 100\}$.

**Example Solutions**   Figure 6 shows some trajectories obtained by 16 agents from a 32-agent population on CVRP100. Unlike TSP, the agent/vehicle performs several tours starting and finishing in the depot.

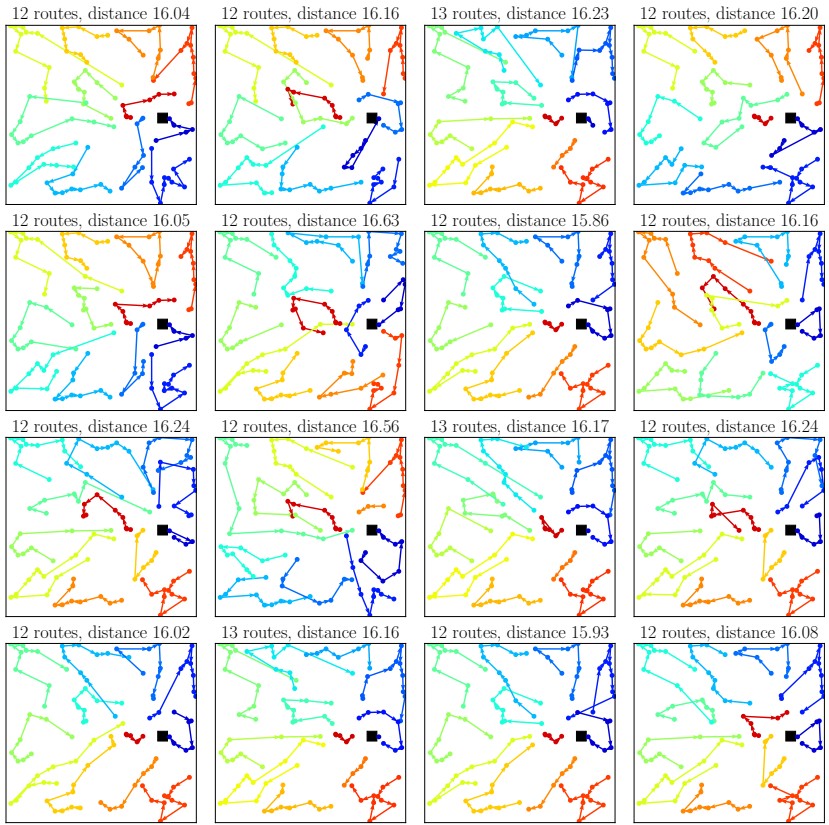

Figure 6: Example CVRP trajectories given by Poppy for 16 agents from a 32-agent population. The depot is displayed as a black square. The edges from/to the depot are omitted for clarity.

### E.3   0-1 Knapsack (KP)

**Problem Description**   Given a set of items, each with a weight and a value, the goal is to determine which items to include in a collection so that the total weight is less than or equal to a given limit and the total value is as large as possible.

**Instance Generation**   Item values and weights are both uniformly sampled in $[0, 1]$. The bag capacity is fixed at 25.

**Training** For KP, and contrary to the other three environments, training an agent is lightning-fast as it only takes a few minutes. In this specific case, we noticed it was not necessary to train a single decoder first. Instead, (i) we directly train a population in parallel from scratch, and (ii) specialize the population exactly as done in the other environments.

**Architecture** We use the same model as in TSP. However, the mask used when decoding is not inverted, and the items that do not fit in the bag are masked together with the items taken so far.

**Hyperparameters** We use the same hyperparameters as in TSP except for the number of starting points $P$ used during training, which we set to 100 after performing a grid-search with $P \in \{20, 50, 100\}$.

### E.4 Job-Shop Scheduling Problem (JSSP)

**Problem Description** We consider the problem formulation described by Zhang et al. [2020] and also used in Bonnet et al. [2023], in the setting of an equal number of machines, jobs and operations per job. A job-shop scheduling problem consists in $N$ jobs that all have $N$ operations that have to be scheduled on $N$ machines. Each operation has to run on a specific machine for a given time. The solution to a problem is a schedule that respects a few constraints:

- for each job, its operations have to be processed/scheduled in order and without overlap between two operations of the same job,
- a machine can only work on one operation at a time,
- once started, an operation must run until completion.

The goal of the problem is to determine a schedule that minimizes the time needed to process all the jobs. The length of the schedule is also known as its makespan.

**Instance Generation** We use the same generation process as Zhang et al. [2020]. For each of the $N$ jobs, we sample $N$ operation durations uniformly in $[1, 99)$ . Each operation is given a random machine to run on by sampling a random permutation of the machine IDs.

**Transition Function** To leverage JAX, we use the environment dynamics implemented in Jumanji [Bonnet et al., 2023] which differs from framing proposed by L2D [Zhang et al., 2020]. However, the two formulations are equivalent, therefore our results on the former would transfer to the latter. Our implementation choice was purely motivated by environment speed.

**Architecture** We use the actor-critic transformer architecture implemented in Jumanji which is composed of an attention layer on the machines' status, an attention layer on the operation durations (with positional encoding) and then two attention layers on the joint sequence of jobs and machines. The network outputs N marginal categorical distributions for all machines, as well as a value for the critic. The actor and critic networks do not share any weights.

**Hyperparameters** Like in Zhang et al. [2020], we evaluate our algorithm with $N = 10$ jobs, operations and machines, i.e. $10 \times 10$ instances. We use REINFORCE with the critic as a baseline (state-value function). Since episodes may take a long time (for an arbitrary policy, the lowest upper bound on the horizon is $98 \times N^3$), we use an episode horizon of 1250 and give a reward penalty of two times the episode horizon when producing a makespan of more than this limit.

## F Time-performance Tradeoff

We present in figure 7 a comparison of the time-performance Pareto front between Poppy and POMO as we vary respectively the population size and the amount of stochastic sampling. Poppy consistently provides better performance for a fixed number of trajectories. Strikingly, in almost every setting, matching Poppy's performance by increasing the number of stochastic samples does not appear tractable.

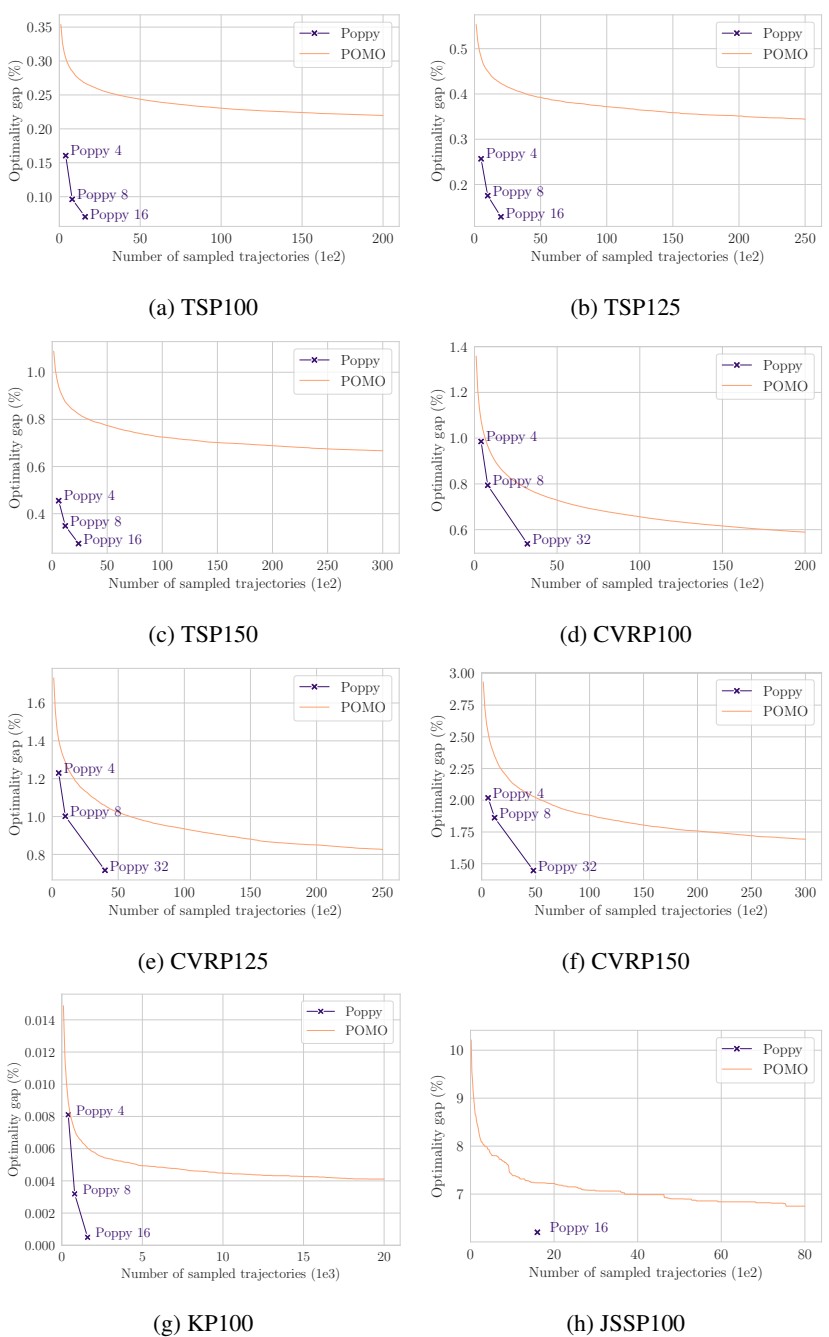

Figure 7: Comparison of the time-performance Pareto front of Poppy and POMO, for each problem used in the paper. The x-axis is the number of trajectories sampled per test instance, while the y-axis is the gap with the optimal solution for TSP, KP and JSSP, and the gap with LKH3 for CVRP.

## G  Larger instances

We display in Table 11 results for POMO and Poppy on much larger TSP instances with 1000 cities. Baselines are taken from Qiu et al. [2023].

Table 11: TSP results (TSP 1000)

| Method | Type | Obj. | Gap | Time |
|---|---|---|---|---|
| Concorde | OR | 23.12 | 0.000% | 6.65H |
| LKH3 | OR | 23.12 | 0.000% | 2.57H |
| EAS (Tab) | RL w/ search | 49.56 | 114% | 63H |
| Dimes | RL w/ search | 23.69 | 2.46% | 5H |
| POMO (16 samples) | RL w/o search | 50.80 | 120% | 39M |
| Poppy 16 | RL w/o search | 39.70 | 71.7% | 39M |

