# OpenReview forum: "Winner Takes It All: Training Performant RL Populations for Combinatorial Optimization"
_NeurIPS.cc/2023/Conference — NeurIPS 2023 poster_

### Official Review · Reviewer_ybFK · 2023-06-26

**Soundness:** 4 excellent
**Presentation:** 4 excellent
**Contribution:** 3 good
**Rating:** 6
**Confidence:** 4

**Summary:**

This paper presents a multi-decoder (population) neural network structure and training method to solve the combinatorial optimization problem. In particular, the paper presents a model update method in which multiple decoders can specialize in different types of problem instances. The method proposed in this paper shows promising results in TSP, CVRP, KP, and JSSP experiments.

**Strengths:**

**S1.** The idea of agent population and the training method used for building it are novel.

**S2.** While the idea is simple and clear, the accuracy improvement by the proposed method is remarkable. When performing inference in a fast and simple form(i.e. without iterative searching for inference), Puppy showed superior results compared to POMO 16 samples or POMO 16 ensemble.

**S3.** The method proposed in the paper is applicable to various types of CO problems.

**S4.** The idea of applying the multi-agent method of this paper to the existing POMO training method is excellent.

**Weaknesses:**

There seems to be no major flaw in the methods presented in the paper. However, there are two opinions related to the TSP/CVRP experiment results.


**W1.** Recent studies that have shown good results in TSP and CVRP experiments are omitted as baselines in Tables 1 and 2, for example [1, 2, 3]. Especially in the case of CVRP experiments, EAS[1], DPDP[2] and SGBS[3] have outperformed LKH3.

**W2.** HGS [4,5] shows better performance than LKH3 as a meta-heuristic algorithm. I recommend that paper authors consider adding HGS as a meta-heuristic baseline in Table 2.


**References**

[1] Andre Hottung, et al. Efficient active search for combinatorial optimization problems. International Conference on Learning Representations, 2022.
[2] Kool, Wouter, et al. Deep policy dynamic programming for vehicle routing problems. Integration of Constraint Programming, Artificial Intelligence, and Operations Research, 2022.
[3] Jinho Choo, et al. Simulation-guided Beam Search for Neural Combinatorial Optimization. Advances in Neural Information Processing Systems 35, 2022.
[4] Thibaut Vidal, et al. A hybrid genetic algorithm for multidepot and periodic vehicle routing problems. Operations Research, 2012.
[5] Thibaut Vidal. Hybrid genetic search for the cvrp: Open-source implementation and swap* neighborhood. Computers & Operations Research, 2022.

**Questions:**

**Q1.** In Algorithm 1, reward of the result of the second best agent $R(\tau_i^{**})$ was used as the baseline(169\~171), but Algorithm 2 seems to have used the POMO shared-baseline of the result of the best agent as the baseline(Algorithm 2, 7\~9). In other words, parameter update by applying the POMO training algorithm only for the best agent for each instance. Is it correct? Please provide a more detailed explanation of the parameter update in Algorithm 2.

**Q2.** How to determine the appropriate number of populations? Are there criteria, considerations, etc. for the decision?

**Q3.** Can you show the change in accuracy, training time, inference time, etc. according to the change in the number of population?

**Q4.** Algorithm 1 Input - Missing 'H' in 'number of training steps H'.

---

> ### Author Rebuttal · Authors · 2023-08-09
>
> We thank the reviewer for the detailed review and appreciating the novelty, simplicity and applicability of our method. We hope our answers address the reviewer’s remaining concerns.
>
> > Q1. In Algorithm 1, reward of the result of the second best agent $R(\tau_{i^{\ast\ast}})$ was used as the baseline(169-171), but Algorithm 2 seems to have used the POMO shared-baseline of the result of the best agent as the baseline(Algorithm 2, 7-9). Please provide a more detailed explanation.
>
> We are grateful to the reviewer for raising this point; indeed we discuss two possible choices of baseline for our training algorithm. In both cases, we rollout our entire population on the problem and train only the best agent, but can consider:
> * POMO baseline: Use the standard shared-baseline introduced in [2] as per Algorithm 2. This is a minimal modification to our re-implementation of POMO and so was initially used for all presented results in the paper.
> * Analytical baseline: As presented in line (169~171); when subsequently deriving the analytical gradient of the population-level objective, we find that the baseline is not the average performance of all agents (i.e. POMO baseline), but rather the performance of the second best agent only.
>
> Given this result, we have since re-run our experiments using the analytical baseline which we are provided below. In summary, we find that in almost all cases the analytical baseline provides better or equal performance than the POMO baseline. We believe that this highlights both that our intuitive “winner takes it all” approach works well even with slightly different choices of baseline, and that indeed our theoretical analysis correctly predicts a modified baseline that can be used in practice with strong performance.
>
> We will include the new results in the appendix of the revised manuscript and add a summary of the above discussion in section 3 of the main text.
>
> Optimal gap (%):
> ### TSP:
>
> | Method/Size | 100  | 125  | 150  |
> |-------------|------|------|------|
> | POMO (16 samples) | 0.27 | 0.42 | 0.81 |
> | Poppy 16 (POMO baseline) | 0.07 | 0.14 | 0.27 |
> | Poppy 16 (analytical baseline) | 0.06 | 0.13 | 0.28 |
>
> ### CVRP:
>
> | Method/Size | 100  | 125  | 150  |
> |-------------|------|------|------|
> | POMO (32 samples) | 0.78 | 1.06 | 2.03 |
> | Poppy 32 (POMO baseline) | 0.52 | 0.71 | 1.43 |
> | Poppy 32 (analytical baseline) | 0.48 | 0.71 | 1.43 |
>
> ### KP:
>
> | Method | Optimal gap |
> |--------|-------|
> | POMO (16 samples) | 0.006 |
> | Poppy 32 (POMO baseline) | 0.0005 |
> | Poppy 32 (analytical baseline) | 0.0001 |
>
> ### JSSP:
>
> | Method | Optimal gap |
> |--------|-------|
> | Single (16 samples) | 7.2 |
> | Poppy 16 (POMO baseline) | 6.2 |
> | Poppy 16 (analytical baseline) | 5.5 |
>
>
> > Q2. How to determine the appropriate number of populations? Are there criteria, considerations, etc. for the decision?
>
> We have experimentally observed that the performance increases with the size of the population (see Tables 1-2, Figure 3). However, larger populations incur higher training and inference costs. We also hypothesize that the population performance could collapse when the population is arbitrarily large, as briefly mentioned in Sec. 5 (lines 308-314). Practically, the primary limits to population size are the availability of compute during the training stage, as well as the inference budget, since populations provide various performance-time trade-offs (Fig 7).
>
> > Q3. Can you show the change in accuracy, training time, inference time, etc. according to the change in the number of population?
>
> We briefly describe the cost of training in TSP with 100 cities in lines 214-216. More precisely, single agent baselines were trained for 5D, 1W, 1D, and 1D for respectively TSP, CVRP, KP and JSSP, while our largest populations were trained until convergence for 4D, 5D, 3D and 3D, which we will add to the paper. Additionally, we want to emphasize that Poppy is already performant with less training budget: Figure 3 (left) outlines the performance of Poppy after just a few hours of training, which is already sufficient for Poppy 4 to reach the performance of POMO (100 samples). The inference times (“Time” column) and accuracies (“Obj.”) for increasing population sizes are studied and provided in Tables 1 (TSP), and 2 (CVRP), and in Fig 7.
>
> > Q4. Algorithm 1 Input - Missing 'H' in 'number of training steps H'.
>
> Thank you, we will fix it in future versions of the paper.
>
> > W1. Recent studies that have shown good results in TSP and CVRP are omitted as baselines (EAS, DPDP and SGBS).
>
> The results presented in the main paper are achieved with pure inference without a search mechanism, which is why we excluded more expensive inference methods like EAS from the main paper. However, we present a comparison to these methods (EAS and DPDP) in Appendix C, referenced on line 322, which we will outline more clearly in the paper. We implement a naive stochastic sampling strategy to give a sense of the performance of Poppy with a larger time budget matching the number of rollouts of EAS. We show that this variant of Poppy can outperform EAS and DPDP even without an explicit adaptation at test time, further highlighting the performance of our approach. We also thank the reviewer for the reference to SGBS, which we will include in future versions.
>
> We would also like to emphasize that EAS and SGBS are search methods used on top of pretrained models, and as such can be considered improvements orthogonal to Poppy. Indeed, in principle, Poppy could be combined with EAS and fine tuned at inference if the inference budget is large enough. We believe that leveraging populations of policies for more efficient inference-time search is a promising direction for future research but is beyond the scope of our current work.
>
> > W2. HGS shows better performance than LKH3.
>
> We thank the reviewer for the reference. We will add it in future versions of the paper.

---

> > ### Comment · Reviewer_ybFK · 2023-08-14
> >
> > I appreciate the authors addressing my concerns with detailed and appropriate responses, as well as conducting additional experiments. However, I will keep my rating for this paper unchanged. I have no further questions.

---

### Official Review · Reviewer_51fw · 2023-07-04

**Soundness:** 2 fair
**Presentation:** 3 good
**Contribution:** 2 fair
**Rating:** 5
**Confidence:** 4

**Summary:**

This paper proposes a construction method that learns a population of agents to improve the exploration of the solution space of combinatorial optimization problems. Experiments demonstrates that the proposed method improves the solving efficiency of four popular NP-hard problems.

**Strengths:**

1. The paper is well-written and easy to follow.
2. Experiments demonstrate that the proposed method achieves state-of-the-art RL results on four popular combinatorial optimization problems.

**Weaknesses:**

1. The motivation is unconvincing. The authors provide a motivating example of a naïve decision-making environment in Figure 1. However, it would be more convincing if the authors could provide an example of combinatorial optimization problems, as this paper aims to develop effective methods to solve combinatorial optimization problems.
2. The idea of learning a population of agents with diverse policies is incremental, as [1] has proposed a similar method.
3. The authors claim that their proposed method can produce a set of complementary policies. However, there is no proof for this claim. Thus, it would be more convincing if the authors could provide theoretical and/or empirical proof for this claim.
4. The experiments are insufficient.
(1) It would be more convincing if the authors could evaluate the proposed method on combinatorial optimization problems with larger sizes, such as TSP with 1000 cities.
(2) Some important baselines are missing. First, the authors may want to compare their method with MDAM [1] on the packing problems. Second, the idea of learning a population of agents is similar to learning an ensemble of agents. Thus, the authors may want to compare their method with baselines with the ensemble learning trick.
(3) It would be better if the authors could provide a time analysis of training cost, as training a population of agents may take much longer time than that of baselines.

[1] Xin, Liang, et al. "Multi-decoder attention model with embedding glimpse for solving vehicle routing problems." Proceedings of the AAAI Conference on Artificial Intelligence. Vol. 35. No. 13. 2021.

**Questions:**

Please refer to Weaknesses for my questions.

**Limitations:**

Yes.

---

> ### Author Rebuttal · Authors · 2023-08-09
>
> We thank the reviewer for their comments. We believe our answers address the concerns raised and hope these enable the Reviewer to reconsider their assessment.
>
> > W1. [The motivation] would be more convincing if the authors could provide an example of combinatorial optimization problems [in Fig. 1].
>
> The motivating example in Figure 1 was chosen because it illustrates the limitations of learning an optimal policy using a single agent: it cannot learn an optimal policy, whereas a population of two agents can. We agree that it is important to relate this to CO problems as these are the focus of our work. However, as we state on lines 154-158, this problem setting can in fact be seen as a toy model for the challenge of solving an NP-hard CO problem in a sequential decision-making setting. Concretely, we posit that the maze prevents the agent from being able to reason over the outcome of its actions, and thus it must learn heuristics over the expected (but uncertain) returns. If we instead consider the first action of a problem such as TSP, the number of possible unique tours that could follow from each action is exponentially large and, for any reasonably finite agent capacity, essentially provides the same obfuscation over the final returns.
>
> We accept that we should have made this link more clear in our manuscript and are happy to extend the discussion of Figure 1 to include a summarized justification.
>
> > W2. The idea of learning a population of agents with diverse policies is incremental, as [1] has proposed a similar method [MDAM].
>
> Whilst MDAM and Poppy both share the intuition of training a population of agents for CO problems, there are substantial differences in both methodology and performance. We provide a comparison in the Related Works (lines 84-92) but to summarize:
> * MDAM trades off performance with diversity by jointly optimizing policies and their KL divergence. Poppy drives diversity by maximizing population-level performance *without* using any explicit diversity metric (which, as we argue in lines 173-175 and 326-328, is a major benefit of our approach).
> * MDAM only drives diversity at the first step of a trajectory since computing the KL divergence for the whole trajectory is intractable. Poppy, by contrast, can generate diverse policies over the entire trajectory.
> * Poppy scales better with population size than MDAM (which requires computing the KL divergence for each pair of agents and thus is only scaled to 5 agents in [1]).
> * Poppy significantly outperforms MDAM in TSP and CVRP (Tables 1 and 2).
>
> In summary, we do not believe that MDAM can be used to detract from Poppy which makes significant contributions both algorithmically (a novel framework for population-based CO) and practically (with significant performance benefits).
>
> > W3. The authors claim that their proposed method can produce a set of complementary policies. However, there is no proof for this claim.
>
> We do provide empirical evidence to support the claim that Poppy can produce a set of complementary policies and will ensure that this is highlighted in revised versions. Specifically:
> * In Fig 3 (right), it can be seen that the *average* agent performance in Poppy 8 has an optimality gap of 1.1%, far from the 0.4% of POMO. However, the performance of the population is 0.09%, outperforming even “POMO (100 samples)” despite using less than 12 times fewer rollouts. This can be interpreted as complementarity: alone, each agent performs worse than POMO, but together they perform far better.
> * Poppy outperforms ensemble methods (Table 1 and 2), showing that it is not reduced to a population of diverse agents.
> * In App. D.1, Fig. 4 (lines 546-557), we show that every agent contributes to the whole population performance.
>
> > W4.1. It would be more convincing if the authors could evaluate the proposed method on CO problems with larger sizes.
>
> Even though this work does not specifically focus on scalability, our method can easily be applied to larger instances since our method is agnostic to the network architecture. Concretely, the Poppy framework can be applied to architectures designed specifically for scalability, e.g. DIMES [1]. However, we use the POMO [2] architecture as it is widely established as the default option in our setting [4, 5, 6] and allows for direct comparison to the SOTA baseline methods. In this context, we scale to the same instance sizes as previous works [2-6].
>
> > W4.2.1.The authors may want to compare their method with MDAM on the packing problems.
>
> The authors of MDAM did not report the performance of their method in KP and JSSP. We also note that on TSP and CVRP, where MDAM reports results, MDAM is not the strongest RL baseline to which we compare: POMO (ensemble) outperforms MDAM in all settings, and is in turn always outperformed by Poppy. Therefore, we believe that it is appropriate to focus on the strongest published baselines for later experiments; and hence use POMO (single and ensemble) for KP and the JSSP-specific RL SOTA method, L2D [3].
>
> > W4.2.2. The authors may want to compare their method with baselines with the ensemble learning trick.
>
> We have indeed performed these experiments, which are denoted by POMO X (ensemble), where X is a number of agents. The results stated in Tables 1, 2, and 3a for TSP, CVRP, and KP, respectively, demonstrate that Poppy outperforms ensembles.
>
> > W4.3. It would be better if the authors could provide a time analysis of training cost.
>
> Single-agent baselines were trained for 5D, 7D, 1D, and 1D for respectively TSP, CVRP, KP and JSSP, while the largest populations we provide were trained until convergence for 4D, 5D, 3D and 3D, which we will add to the paper. Additionally, we want to emphasize that Poppy is already performant with less training budget: Figure 3 (left) outlines the performance of Poppy after just a few hours of training, which is already sufficient for Poppy 4 to reach the performance of POMO (100 samples).

---

> > ### Comment · Area_Chair_YoAd · 2023-08-16
> > **AC note: Please engage with authors**
> >
> > Hi Reviewer 51fw,
> > Please engage with the authors. They have put in a significant amount of effort to respond to the concerns, and improve their work. I'd encourage you to respond asap and give the authors an opportunity to continue improving their work!

---

> > > ### Comment · Reviewer_51fw · 2023-08-17
> > > **Thanks for the kind reminder**
> > >
> > > Dear AC
> > > Thank you for the kind reminder. I have read the authors' rebuttal and the other reviews. I am engaging with authors now.

---

> > > > ### Author Response · Authors · 2023-08-17
> > > >
> > > > We thank the reviewer for the continued discussion.  We regret that concerns still remain, however we remain confident that these points can be satisfactorily addressed and do not represent significant issues with our work.
> > > >
> > > > Whilst we hoped our original responses would clarify these points, we are more than happy to address them further.  However, **we would kindly ask that the Reviewer clarify specifically why they do not feel our responses were sufficient** such that we can continue to address remaining questions.
> > > >
> > > > ### Concern 1
> > > >
> > > > We are uncertain about the Reviewer's specific concerns. Is the question centered on the rationale for having a diverse policy set rather than a single policy when addressing NP-hard CO problems? Or does the Reviewer find the toy example in Figure 1 overly simplistic to illustrate this point?  If it is the latter, we are open to substituting the maze with a more intricate routing problem. The overarching argument remains valid, should the Reviewer prefer this change.
> > > >
> > > > However, we wish to highlight that Poppy's robust empirical performance is our principal contribution. Our goal with Figure 1 and the related discussion was to elucidate the intuition behind proposing the Poppy objective in a simplified setting. Even if there are suggestions for alternative motivating examples or potential for deeper theoretical insights, we maintain that Poppy's demonstrated effectiveness justifies our contribution.
> > > >
> > > > ### Concern 3
> > > >
> > > > We are confident that our results demonstrate that Poppy produces a set of complementary policies. We'll outline the evidence, and if the reviewer remains unconvinced, we'd value specific feedback.  Should our manuscript lack clarity on this topic, we're open to making necessary revisions.
> > > >
> > > > By "a complementary set of policies", we mean:
> > > > 1. Individual policies in the population are most performant on different sub-distributions of problem instances.
> > > > 2. As a collective, the policies encompass the entire problem instance distribution, ensuring that the performance of the best policy for any sampled problem instance remains strong.
> > > >
> > > > **Evidence for (1)**
> > > > Removing any agent from the population reduces the overall population-level performance, as seen in Fig 4's bottom-right (Appendix D.1). This signifies that every agent is optimal for a specific sub-distribution of problems. It's important to note that these contributions are computed with respect to greedy rollouts of each policy, and so cannot be attributed to stochasticity in the agent performance.
> > > > Fig. 3 illustrates the effects of specialization. As the population size grows, the expected performance of a randomly selected agent on a random problem gets worse. However, the performance of the best-suited agent for a random problem is improved. This highlights that when each agent has a narrower target sub-distribution, it learns an even more specialized policy, which is even better (worse) for problem instances in (out) of the target sub-distribution. This is in contrast with a simple policy ensemble, for which the average agent performance would remain the same regardless of the population size.
> > > >
> > > > **Evidence for (2)**
> > > > In a head-to-head comparison of Poppy's collective performance against POMO – either sampling a single POMO agent N times, or training an ensemble of N POMO agents – Poppy consistently surpasses both benchmarks at the same cost (Tables 1, 2 and 3). This demonstrates that the set of policies learned by Poppy, as a collective, excels across all problem instances.
> > > >
> > > >
> > > > ### Concern 4
> > > >
> > > > **4.1**: As described in our original response, we agree that scaling to extremely large CO problems is an important challenge but note that it is a distinct research topic.   Baseline methods to which Poppy is compared (POMO, EAS, MDAM, LIH, NeuRewriter, NLNS, 2-Opt-DL, CVAE-Opt, DPDP) tackle the same order of problem sizes as our work.  This ensures fair comparison and our leading performance in these problem settings (see Tables 1, 2 and 3) is not questioned by any reviewers.  If the Reviewer’s belief is that scaling up to TSP1000 is mandatory for all research in this domain, we would respectfully disagree and note this is a standard not applied to prior works.
> > > >
> > > > **4.2.1**: The requested MDAM baseline for KP and JSSP is shown to be significantly outperformed by other RL methods on TSP and CVRP (Tables 1 and 2). Given the absence of published results for MDAM on these problems and the impracticality of re-implementing all existing neural CO methods, we prioritized the strongest baselines: POMO+sampling and POMO+ensembling. If the Reviewer does not agree this is reasonable we would welcome specific inquiries.
> > > >
> > > > **4.2.2**: The requested baseline was included in our original submission, so we believe this issue should be considered resolved.

---

> > ### Comment · Reviewer_51fw · 2023-08-17
> > **Thanks for the authors’ rebuttal**
> >
> > Thanks for the authors' response. However, my major concerns 1, 3, and 4 have not been properly addressed. Thus, I would lean toward rejection given the current status of my communication with the authors.
> >
> > 1. (Concern 1 in weaknesses) The authors do not provide a convincing example for the motivation.
> >
> > 2. (Concern 3 in weaknesses) It is unclear to me why Fig 3 demonstrates that Poppy can produce a set of complementary policies.
> >
> > 2. (Concern 4 in weaknesses) My concerns 4.1 and 4.2 have not been properly addressed.

---

### Official Review · Reviewer_wqBo · 2023-07-06

**Soundness:** 4 excellent
**Presentation:** 3 good
**Contribution:** 3 good
**Rating:** 8
**Confidence:** 4

**Summary:**

The paper proposes a new training procedure that allows to train a diverse set of policies for solving combinatorial optimization problems. Most existing approaches for these problems train a single policy/agent and aim to construct a solution in a single shot (or by sampling multiple solutions). In contrast, the authors propose to use a population of agents that is trained such that a diverse set of solutions can be obtained by solving a problem instance by each population member. To this end, the paper proposes a new training objective that aims to increase the maximum reward over all K population members. This is implemented by backpropagating only through the best performing agent for each instance during the training phase. Experimentally, the authors show that this approach is indeed able to learn a set/population of complementary policies. Furthermore, the results demonstrate that the performance of a population trained via the proposed approach is clearly superior to a population in which all agents are trained independently in standard-fashion.

**Strengths:**

- While other papers have proposed to train a population of agents, the proposed training procedure (that always only trains the best agent of the population per instance) is novel.
- The proposed methods show a significant performance improvement on the considered problems. For example, on the traveling salesperson problem the method finds solutions with a gap to optimality of 0.07% (in comparison to a gap of 0.27% of the state-of-the-art POMO method). To the best of my knowledge, the proposed method is currently the best neural combinatorial optimization method for quick solution generation (i.e., without a search component).
- The method succeeds in training a population of complementary agents. The authors show experimentally that the average solution quality decreases during the training while the best performance over all agents increases. This means that the agents successfully specialize on specific strategies.
- The authors evaluate their method on 4 different combinatorial optimization problems which demonstrates the wide applicability of the technique. On all problems, the method significantly improves the performance.
- The considered problem of learning heuristics for combinatorial optimization problems has gotten a lot of attention in the literature and is a very promising research area.
- The proposed method is simple and can be easily applied to other neural construction methods. Thus, it is likely that other researchers will use the proposed technique in their work.
- Overall, the paper is well written and clearly organized.


**Weaknesses:**

- Overall, the additional training of a population with the proposed method is quite resource intensive because only one of the K rollouts is used for backpropagation. In some settings, the additional resources needed for the population training phase might not be worth the obtained performance improvements.

**Questions:**



**Limitations:**

- The paper does not discuss limitations of the proposed method.

---

> ### Author Rebuttal · Authors · 2023-08-09
>
> We thank the reviewer for their positive comments and outlining the strengths of our work (novelty, performance, applicability, clarity). While there are no questions, we are happy to provide some feedback on a comment made in the review.
>
> > Overall, the additional training of a population with the proposed method is quite resource intensive because only one of the K rollouts is used for backpropagation. In some settings, the additional resources needed for the population training phase might not be worth the obtained performance improvements.
>
> We agree with the reviewer that the training process can be intensive. However, we still would like to emphasize that the practical cost of training the population is still in the order of training a single agent (for example, Poppy 16 costs 80% of the training of the initial single agent, as we briefly mention on lines 214-215). Crucially, the cost of training the population is substantially reduced by cloning the final few layers of a pre-trained single agent to initialize the population and then fine-tuning each agent. Future works could consider strategies that enable the early detection of the poor performing agents and hence avoid full rollouts for them.

---

> > ### Comment · Reviewer_wqBo · 2023-08-21
> >
> > Thank you for your response. I keep my very high rating and believe that the paper should be accepted.

---

### Official Review · Reviewer_Ym2T · 2023-07-26

**Soundness:** 3 good
**Presentation:** 4 excellent
**Contribution:** 3 good
**Rating:** 5
**Confidence:** 4

**Summary:**

The paper proposes that populations of agents can produce better results than using single agents. This leads to a new "Poppy" algorithm which performs policy gradient updates only on the agent which produced the highest reward. This is then applied to combinatorial optimization problems using attention-based architectures to demonstrate fast and competitive results.

**Strengths:**

* The proposed "Poppy" algorithm appears to be a novel policy gradient variant. I do wonder however, if such an algorithm can be extended to general RL settings, rather than only combinatorial optimization.
* Overall the paper is a clean read and presentation is strong. I did not have any issues with understanding.
* Experimental results are reasonable. The method demonstrates good performance and low inference costs.


**Weaknesses:**

The core issue of the proposed method, is that there is no explicit "diversity" objective being optimized, which makes this a theoretical weakness. However, the paper does defend against this through multiple explanations (e.g. agents will tend to specialize) and empirical evidence (e.g. best agent improves which population worsens).





**Questions:**

* Could the proposed Poppy RL algorithm be more generally applicable to any MDP? If so, which problems (outside of combinatorial optimization) do you think can be most benefited?
* Following up on my weakness section - In which types of MDPs/cases could "diversity" behavior not appear from Poppy training, since diversity is not an explicitly optimized objective?

**Limitations:**

Limitations section appears comprehensive. No issues.

---

> ### Author Rebuttal · Authors · 2023-08-09
>
> We thank the reviewer for their comments and appreciating the presentation and clarity of the work. We hope our answers address all concerns and are sufficient to reconsider the assessment.
>
> > Could the proposed Poppy RL algorithm be more generally applicable to any MDP? If so, which problems (outside of combinatorial optimization) do you think can be most benefited?
>
> The Poppy framework (cloning a single agent, then applying the Poppy objective) is applicable to any model architecture. However, Poppy only makes sense in settings where (i) there is a distribution of problem instances and (ii) these instances can be attempted multiple times at inference (e.g., self-driving cars would not fit into this category). A lot of environments have these two features, including combinatorial problems, text/code generation problems or even theorem proving. Moreover, environments which are based on a simulator that we can repeatedly use satisfy (ii), as several attempts can be performed virtually before acting.
>
> We make a brief reference to this question in lines 324-326, however, we agree that this is a natural question and will extend this to include the above discussion in the final version of the manuscript.
>
> > In which types of MDPs/cases could "diversity" behavior not appear from Poppy training, since diversity is not an explicitly optimized objective?
>
> We note that diversity is not the aim of Poppy; rather we aim to maximize performance - therefore we expect diversity to emerge if, and only if, it increases the population-level performance (i.e. taking the best result from any agent in the population on each sampled problem instance). Correspondingly, if only a single policy can achieve the highest practical returns across all problem instances we would expect the population to collapse to have only one agent providing all of the performance (as the other agents are never the best on a sampled problem and thus never get trained). Trivial examples would be when the training distribution consists of only a single problem instance, or when a single simple strategy (that can be exactly learned by an agent) can reach optimal performance on all problem instances (e.g. if we train on only TSP problem instances where the shortest tour is obtained by always moving to the nearest unvisited city).
>
> > The core issue of the proposed method, is that there is no explicit "diversity" objective being optimized
>
> We firmly believe that this is, in fact, a key strength of our approach, for the following reasons:
> * Population diversity is only a proxy for population performance. Poppy directly optimizes the target metrics and attains diversity as a by-product, which is more aligned with our objective of solving CO problems.
> * Diversity is tricky to measure and thus to optimize. The KL divergence has been used in CO (MDAM [6]), but it is challenging to scale as the complexity grows as the size of the population squared, and was only applied to the very first action, which limits the performance (as we argue on L85-L92). Another approach in the line of quality-diversity methods would be to encode explicit behavioral markers. However, this can be difficult to implement in the case of CO problems where there are no canonical behavioral markers, and so these would have to be handcrafted for each specific problem.
> * Even if we were to suppose that we have a way to measure and optimize diversity, it is not clear how to balance the RL and diversity objective. Indeed, different problems may necessitate various degrees of diversity and too much diversity can hurt final performance: this implies tuning the hyperparameter used in the loss trade-off, as done in [6].
> * We demonstrate empirically that optimizing our objective produces a set of diverse agents (e.g., see Figures 4-6 in the Appendix) across several domains *without* handcrafting any problem-specific notions of diversity.

---

> > ### Comment · Area_Chair_YoAd · 2023-08-16
> > **AC Note: Please engage with the authors**
> >
> > Hi Reviewer Ym2T, Please engage with the authors. They have put in a significant amount of effort to respond to the concerns, and improve their work. I'd encourage you to respond asap and give the authors an opportunity to continue improving their work!

---

> > ### Comment · Reviewer_Ym2T · 2023-08-16
> > **Keeping current score of acceptance**
> >
> > I thank the authors for their time writing their response. I do think the paper has adequately defended its position on the diversity issue, and has achieved good results.
> >
> > The proposed method indeed is interesting, and it appears to be a more general technique than just for combinatorial optimization (although its generality is not explored in this paper). At the moment, the paper focuses mostly on combinatorial optimization cases however, and so the impact of the paper may be limited.
> >
> > Thus I am fine with lukewarm acceptance at the moment.

---

> > > ### Author Response · Authors · 2023-08-17
> > >
> > > We appreciate the Reviewer's feedback and are thankful our work's merit isn't in doubt.
> > >
> > > The Reviewer comments on a possible narrow appeal due to our focus on CO problems. We respectfully differ in this view. Combinatorial Optimization is a significant and long-standing research area, with related papers frequently presented at top venues like NeurIPS. For reference, we've listed several recent NeurIPS publications on neural CO at the end of our response.
> > >
> > > To further address the Reviewer’s concern, we're open to expanding our introduction and related work sections to underscore real-world applications of CO problems, such as routing, scheduling, and chip placement.
> > >
> > > However, we fully appreciate that gauging the impact of specific topics can be subjective and simply wished to take this chance to share our perspective. We're grateful for the reviewer's constructive feedback and their positive evaluation of our research.
> > >
> > > **NeurIPS 2022**
> > > * Malherbe et al. “Optimistic Tree Searches for Combinatorial Black-Box Optimization”.
> > > * Choo et al. “Simulation-guided Beam Search for Neural Combinatorial Optimization”.
> > > * Kim et al. “Sym-NCO: Leveraging Symmetricity for Neural Combinatorial Optimization”.
> > > * Wang et al. “Unsupervised Learning for Combinatorial Optimization with Principled Objective Relaxation”.
> > >
> > > **NeurIPS 2021**
> > > * Kim et al. “Learning Collaborative Policies to Solve NP-hard Routing Problems”.
> > > * Ma et al. “Learning to Iteratively Solve Routing Problems with Dual-Aspect Collaborative Transformer”.
> > > * Wang et al. “A Bi-Level Framework for Learning to Solve Combinatorial Optimization on Graphs”.
> > > * Kwon et al. “Matrix encoding networks for neural combinatorial optimization”.

---

### Author Rebuttal · Authors · 2023-08-09

We thank the reviewers for their detailed feedback on our manuscript. We summarize some common points across reviewers:

* *Not having an explicit objective for diversity is an advantage, not a weakness.* We empirically show that Poppy attains diversity as a by-product of optimizing the proposed population objective. Crucially, Poppy does not require any problem-specific notion of diversity, which makes it generally applicable to a wide range of problems while still achieving SOTA performance across them. See answers to Reviewers Ym2T and 51fw for more details.
* *The cost incurred by training the population is comparable to that of training a single agent.* See answers to Reviewers wqBo, 51fw and ybFK for details.

In the future, we will *update* the paper in response to the reviewers’ comments. The main changes include:
* Include the results reported in the answer to Reviewer ybFK.
* Make the link between Fig. 1 (motivation for populations) and CO problems clearer.
* Clarify why the policies learned by the agents are complementary using Fig 3.
* Explain why Poppy is potentially applicable to problems other than CO.
* Include the solver HGS as a baseline for CVRP.
* Include SGBS as a baseline in Appendix C.

Additionally, we provide below several references used in the answers provided to the reviewers:


[1] Qiu et al. 2022. DIMES: A Differentiable Meta Solver for Combinatorial Optimization Problems.
[2] Kwon et al. 2020. POMO: Policy Optimization with Multiple Optima for Reinforcement Learning.
[3] Zhang et al. 2020. Learning to Dispatch for Job Shop Scheduling via Deep Reinforcement Learning.
[4] Hottung et al. 2022. Efficient Active Search for Combinatorial Optimization Problems.
[5] Kool et al. 2019. Attention, Learn to Solve Routing Problems!
[6] Xin et al. 2021. Multi-decoder attention model with embedding glimpse for solving vehicle routing problems.

---

### Decision · Program_Chairs · 2023-09-21

**Decision:**

Accept (poster)

**Comment:**

The authors propose a population based training objective to use RL for solving combinatorial optimization problems. These techniques aim to learn policies that specialize and are diverse in the solution space. They show applicability to problems like TSP, CVRP, KP and JSSP. The reviewers mostly appreciated the motivation and methodology of the work and largely recommend an accept.

It is important that you address their concerns in the final version:
1. Better example for motivation
2. Experiments on combinatorial optimization problems with larger sizes, such as TSP with 1000 cities.
3. Some important baselines are missing - eg MDAM [1] on the packing problems.

The paper has interesting ideas to discuss, but addressing these 3 concerns would be very good for the camera ready.